# Structural basis for assembly and disassembly of the IGF/IGFBP/ALS ternary complex

Hyojin Kim[1,3], Yaoyao Fu[2,3], Ho Jeong Hong[2], Seong-Gyu Lee[2], Dong Sun Lee[2] & Ho Min Kim [1,2] ✉

Insulin-like growth factors (IGFs) have pleiotropic roles in embryonic and postnatal growth and differentiation. Most serum IGFs are bound in a ternary complex with IGF-binding protein 3 (IGFBP3) and acid-labile subunit (ALS), extending the serum half-life of IGFs and regulating their availability. Here, we report cryo-EM structure of the human IGF1/IGFBP3/ALS ternary complex, revealing the detailed architecture of a parachute-like ternary complex and crucial determinants for their sequential and specific assembly. In vitro biochemical studies show that proteolysis at the central linker domain of IGFBP3 induces release of its C-terminal domain rather than IGF1 release from the ternary complex, yielding an intermediate complex that enhances IGF1 bioavailability. Our results provide mechanistic insight into IGF/IGFBP3/ALS ternary complex assembly and its disassembly upon proteolysis for IGF bioavailability, suggesting a structural basis for human diseases associated with *IGF1* and *IGFALS* gene mutations such as complete ALS deficiency (ACLSD) and IGF1 deficiency.

Insulin-like growth factors (IGFs: IGF1 and IGF2) are members of the insulin superfamily of hormones, which have pleiotropic roles in embryonic and postnatal growth and differentiation through activating IGF receptors (IGFRs: IGF1R and IGF2R) and the insulin receptor signaling cascade[1]. In the circulation, IGFs can form binary complexes (10%–15%) with IGF-binding proteins (IGFBPs: IGFBP1, IGFBP2, IGFBP4, or IGFBP6), or ternary complexes (80%–90%) with IGFBPs (IGFBP3 or IGFBP5) and acid-labile subunit (ALS)[2–4]. Free IGFs in serum have a half-life of less than 10 min, but a binary complex increases this half-life to 30–90 min, whereas a ternary complex maintains it up to 16–24 h[5]. These imply that the binary (IGF/IGFBP) and ternary complexes (IGF/IGFBP/ALS) transport IGFs, protect them from degradation for a prolonged half-life, and limit their binding to IGFRs. As the IGF axis plays a crucial role in physiological processes, including cell growth, metabolism, and differentiation, alterations in IGF expression levels and genetic mutations are implicated in various pathological conditions

including growth retardation, diabetes, osteoporosis, neurodegenerative diseases, obesity, and cancer[6–8].

Despite high sequence similarity (~50%) between IGFs and insulin, IGFs are single-chain polypeptides consisting of B, C, A, and D domains (Fig. 1a), whereas the C-peptide of pro-insulin is excised and two processed chains (A and B chains) are connected by disulfide bonds to form mature insulin[9]. As IGFs can be released after IGFBP proteolysis[10], their bioavailability in circulation and interaction with receptors are stringently regulated by IGFBP-degrading proteases. Six members of the human IGFBP family (IGFBP1–6) consist of N-terminal, central linker, and C-terminal domains. The central linker domain (CLD), a major proteolytic site for IGF bioavailability, is unique to each IGFBP, but their cysteine-rich N-terminal domain (NBP) and C-terminal domain (CBP) are highly conserved. Previous structural studies of IGF1 in complex with IGFBP [miniNBP5 (PDB ID: 1H59), NBP4/CBP4 (PDB ID: 2DSR), hybrid NBP4/CBP1 (PDB ID: 2DSQ)] reveal that IGF1 is wedged

[1]Graduate School of Medical Science and Engineering, Korea Advanced Institute of Science and Technology (KAIST), Daejeon 34141, Republic of Korea. [2]Center for Biomolecular & Cellular Structure, IBS, Daejeon 34126, Republic of Korea. [3]These authors contributed equally: Hyojin Kim, Yaoyao Fu. ✉e-mail: hm_kim@kaist.ac.kr

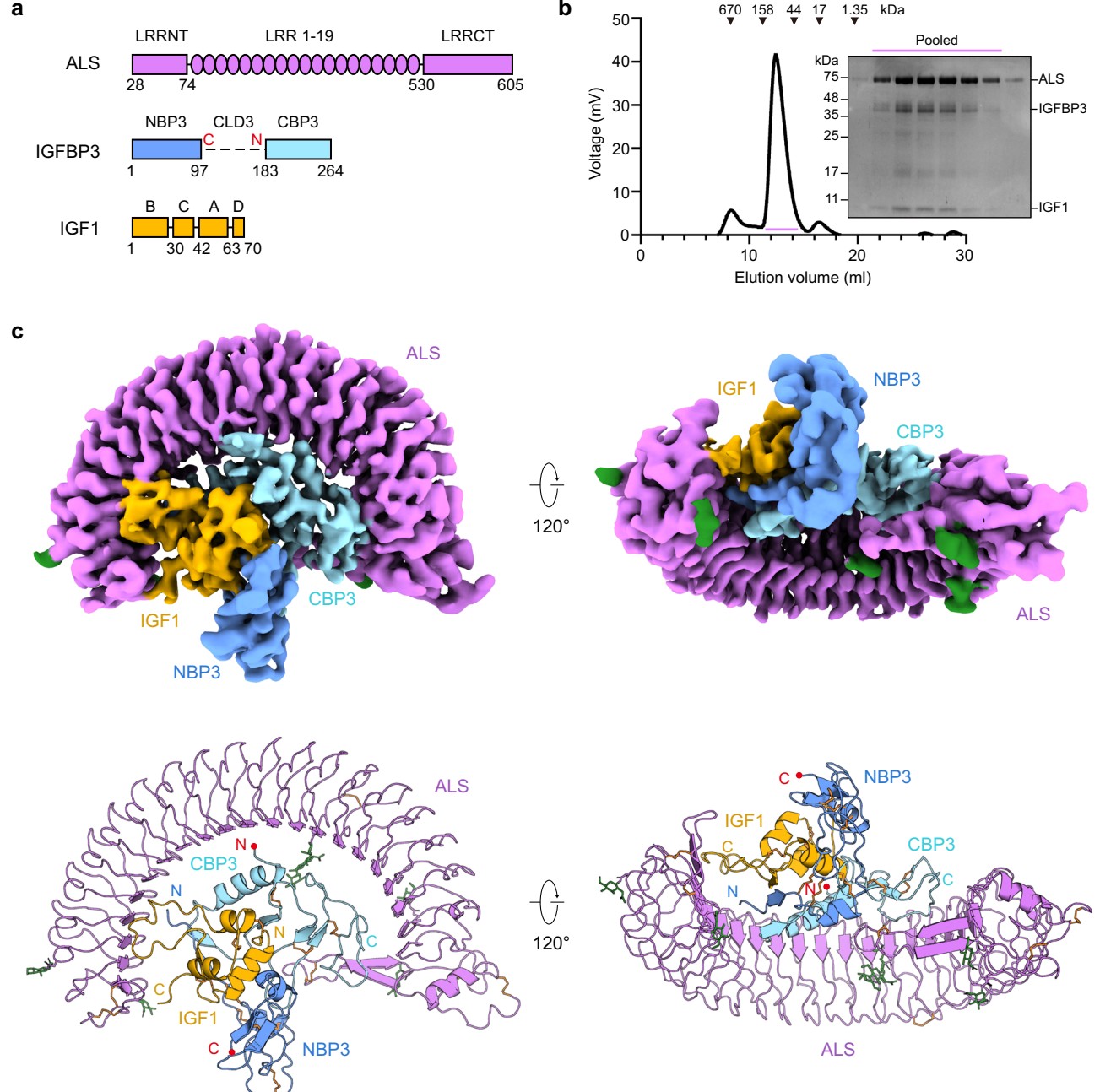

**Fig. 1 | Overall structure of human IGF1/IGFBP3/ALS ternary complex.**
**a** Schematic diagram of human ALS, IGFBP3, and IGF1. Each domain is labeled above the diagram, and the amino acids are numbered under the diagram. In IGFBP3, the central linker domain, which was not resolved in the cryo-EM map, is shown as a dashed line. The boundary of the central linker domain is labeled with a red C and N, indicating the C-terminus of NBP3 and N-terminus of CBP3, respectively. **b** Size-exclusion chromatography (SEC) profile of human IGF1/IGFBP3/ALS ternary complex (left) and SDS-PAGE analysis of the SEC peak fractions stained with Coomassie blue (right). Pooled fractions for cryo-EM analysis are marked with the bar (violet). Similar results were observed in three independent experiments. **c** Overall structure of the human IGF1/IGFBP3/ALS ternary complex. Cryo-EM maps (top) and the corresponding cartoon models (bottom) of the ternary complex are shown. ALS, N-terminal domain of IGFBP3 (NBP3), C-terminal domain of IGFBP3 (CBP3), and IGF1 are in violet, blue, cyan, and yellow, respectively. The C-terminal end of NBP3 and N-terminal end of CBP3 are marked with red dots. Disulfide bridges (orange) and N-linked glycans (green) are shown as sticks.

into the cleft formed by the palm/thumb-like NBP and the flat-shaped CBP[11,12]. The structure of the IGF1/IGF1R complex explains how only one IGF1 molecule binds the Γ-shaped asymmetric IGF1R dimer for IGF1R activation and why IGF1 with longer C-domain loops than IGF2 exhibits stronger binding to IGF1R[13]. Moreover, these structures also suggest that the IGF1 binding pocket in IGF1R is too small to accommodate the binary complex and that the IGF residues involved in high-affinity binding to the IGF1R primary site are engaged in interaction with IGFBP, thereby preventing binding of the IGF/IGFBP binary complex to the receptor. However, these structures provide limited information about the flexible regions of the binary complex, including the N-terminal thumb loop, CLD, and the C-terminal end of IGFBPs, or about the C-domain loop of IGF1 that may be involved in ternary complex formation or IGF release from the ternary complex.

ALS is produced almost exclusively by the liver and secreted into the circulation. It belongs to the leucine-rich repeat (LRR) superfamily consisting of 19 LRR motifs flanked by LRR N- and C-terminal modules (LRRNT and LRRCT, respectively)[14] (Fig. 1a). Individuals with complete

ALS deficiency (ACLSD, OMIM #615961) have mutations in the *IGFALS* gene and exhibit severely reduced serum IGF1 and IGFBPs (particularly IGFBP3), leading to low birth weight and length, reduced head circumference and height, pubertal delay, and insulin resistance[15–17]. These outcomes indicate that ALS is crucial for maintaining the integrity of the circulating IGF/IGFBP system and that the IGF/IGFBP/ALS ternary complex serves as an IGF reservoir facilitating the endocrine actions of IGFs. Of interest, ALS has no affinity for free IGFs or only a low affinity for apo IGFBP3 and IGFBP5[18]. The central electronegative surfaces together with the N-linked carbohydrates on ALS are proposed to be crucial in its interaction with the IGF/IGFBP binary complex[19,20]. The lack of a defined structure for IGF/IGFBP/ALS ternary complexes, however, has hindered understanding of the structural organization underlying a prolonged IGF half-life, as well as the distinct affinity of IGF/IGFBPs (IGFBP3 and IGFBP5) for ALS binding.

Here, we determine the cryo-electron microscopy (cryo-EM) structure of the human IGF1/IGFBP3/ALS ternary complex at an overall resolution of 3.2 Å. Our structure reveals that the IGF1/IGFBP3 binary complex engages in long-range interactions with the entire concave surface of ALS. Also, interacting with the IGF1/IGFBP3 binary complex are the hook loop at LRRCT of ALS and the N-linked glycans attached to N368 of ALS. Our structural and biochemical analysis clearly shows that the N-terminal thumb of NBP3 as well as the α3 helix and C-terminal loop of CBP3 are key determinants for assembly of ALS with specific binary complexes (IGF/IGFBP3 or IGF/IGFBP5). Our results also indicate that the CLD of IGFBP3 sterically blocks ALS binding unless IGFBP3 binds to IGF1, and that proteolysis of IGFBP3 CLD in the ternary complex induces release of CBP3 rather than IGF1 from the ternary complex. Taken together, our findings provide mechanistic insight into assembly of the IGF/IGFBP3/ALS ternary complex and its disassembly for IGF bioavailability and allow us to interpret the structural effects of IGF1 and ALS mutations in ACLSD and IGF1 deficiency (OMIM #608747).

## Results

### Overall structure of the human IGF1/IGFBP3/ALS ternary complex

Using HEK293F cells, we co-expressed full-length human IGF1, IGFBP3, and ALS, and a stable ternary complex was co-purified by affinity chromatography and subsequent size-exclusion chromatography (SEC) (Fig. 1b). We determined the cryo-EM structure of the human IGF1/IGFBP3/ALS ternary complex at an overall resolution of 3.2 Å (Fig. 1c and Supplementary Figs. 1–2, and Supplementary Table 1). The atomic models of ALS were constructed de novo, and IGF1/IGFBP3 models were manually built after fitting the crystal structure of the IGF1/IGFBP4 complex (NBP4 and CBP4 fragment without CLD; PDB: 2DSR)[12] on the cryo-EM map. The glycan densities for five N-glycosylation sites of ALS (N64, N96, N368, N515, and N580) were clearly visible in the cryo-EM map (Supplementary Fig. 2c). The overall structure of the ternary complex had a "parachute" shape with 1:1:1 stoichiometry (IGF1:IGFBP3:ALS). The IGF1/IGFBP3 binary complex bound to almost the entire concave surface of the horseshoe-like ALS (Fig. 1c), and both IGF1 and IGFBP3 participated in the interaction with ALS. Consistent with the previously reported structure of IGF1 in complex with NBP4 and CBP4 fragments[12], IGF1 was clamped by the NBP3 and CBP3 of full-length IGFBP3. Although the densities for IGFBP3 CLD (CLD3) were not well resolved in the cryo-EM map because of its structural flexibility, extra density around IGF1 under a lower threshold indicated that the CLD3 might act as a "mechanical flap" covering IGF1 not yet wrapped by NBP3 and CBP3 (Supplementary Fig. 2e).

### Human ALS structure

Human ALS, which belongs to the typical LRR family, consists of 19 LRR motifs with a consensus sequence (xLxxLxLxxNxLxxLxxxxFxx/Lx)

flanked by LRRNT and LRRCT, and had a curved solenoid structure (Fig. 2a). A total of 21 parallel β-strands [LRRNT (βN), 19 LRRs (β1–19), and LRRCT (βC)] on the concave surface had uniform twist angles and radii throughout the entire curvature, giving ALS a flat horseshoe-like structure. The convex surface of the LRR motifs was formed by well-organized loops except for LRR18, which contained a 3₁₀ helix, and the unique disulfide bond (C373–C397) in the LRR motif stabilized the convex portion of LRR13 and LRR14. The conserved phenylalanine-spine and asparagine-ladder (except LRR2, LRR15, and LRR17–19 for the phenylalanine-spine, and LRR2 and LRR14 for the asparagine-ladder) were directed to the interior of the solenoid core (Fig. 2a). The LRRNT (residues 28–74) and LRRCT (residues 530–605) were stabilized by disulfide bonds: C41–C47/C45–C60 and C566–C571/ C540–C583/C542–C605, respectively. Five N-linked glycosylation sites of ALS were distributed on LRRNT (N64), the LRR motif (N96 at the convex surface, N368 at the concave surface, and N515 at the ascending flank), and LRRCT (N580).

Of note, a long loop (residues 559–580) of LRRCT and N-linked glycans at N368 extended from the concave surface, protruding into the central axis of the ALS curvature (Fig. 2a, b) (hereafter, we refer to this extended loop in LRRCT as a "hook loop"). The tip and stem of the hook loop were stabilized by a disulfide bridge (C566–C571) and antiparallel β-sheet (residues 560–564 and 575–579), respectively. The bottom of the hook loop was anchored to the LRRCT. In particular, R560 at the beginning of the hook loop was tethered to D593, R595, and D596 by forming an elaborate electrostatic network, and N579 at the end of the hook loop formed hydrogen bonds with R595 and the main chain of L531 on LRRCT. In addition, the aromatic rings of F561, Y576, and Y578 and N-acetylglucosamine (NAG) attached to N580 were stacked with each other below the β-sheet of the hook loop (β CH1 and β CH2), and W532, P559, and V562 formed the hydrophobic networks above the β-sheet (Fig. 2b and Supplementary Figs. 3a and 4a). These extensive interactions conferred a stable and rigid structure on the hook loop, despite its 26 Å length. The significance of hook loop and glycan at N368 for interactions with the IGF1/IGFBP3 complex is described in a later section.

### Structure of full-length IGFBP3 in complex with IGF1

The IGF1/full-length IGFBP3 complex within the ternary complex adopts similar global folds with the human IGF1/IGFBP4 complex (NBP4 and CBP4 fragment without CLD; PDB ID: 2DSR), with a backbone root mean square deviation (RMSD) of 1.189 Å (Fig. 2c). Our cryo-EM structure clearly defined highly conserved disulfide bonds (3 in IGF1, 6 in NBP3, and 3 in CBP3) crucial for rigidity of the core modules. The IGF1 B domain α helix (A8-V17) was wedged deep into the cleft formed by an edge of the NBP3 palm and a flat surface of CBP3 (Fig. 2d; Supplementary Figs. 3 and 4b). The interactions between IGF1 and IGFBP3 were mainly mediated by hydrophobic contact. Residues (L5, A13, F16, L54, and L57) on the IGF1 N-terminal loop (G1-L5) and one face of the IGF1 B domain α helix and IGF1 A domain α2 helix (L54-Y60) constituted long-range hydrophobic networks with residues on NBP3 (P38, C54, I56, C67, L77, L80, and L81) (Fig. 2d). V11 on the other face of IGF1 B domain α helix also engaged in hydrophobic interactions with V208, I210, and P226 of CBP3 (Fig. 2d). At the entrance of the cleft, other hydrophobic residues of IGF1 (V44 and F49 on the A domain α1 helix) were bound to the hydrophobic flat surface of CBP3 consisting of C186, M190, L194, I210, and C213 (Fig. 2e; Supplementary Figs. 3 and 4b). These hydrophobic interactions were further tightened by electrostatic interactions [N212 (CBP3)-E9 (IGF1); K215 (CBP3)-T4 (IGF1); S227 (CBP3)-Q15 (IGF1)]. It is worth noting that D12 on the IGF1 B domain α helix made direct contact with both NBP3 Y57 and CBP3 S227. For NBP3 and CBP3 interaction, a hairpin-like loop of CBP3 between C224 and C235 made direct contacts with NBP3 (Fig. 2e; Supplementary Figs. 3 and 4c). Similar to IGFBP4, the side chain of R232 on CBP3 stacked with the aromatic ring of conserved Y57 (NBP3).

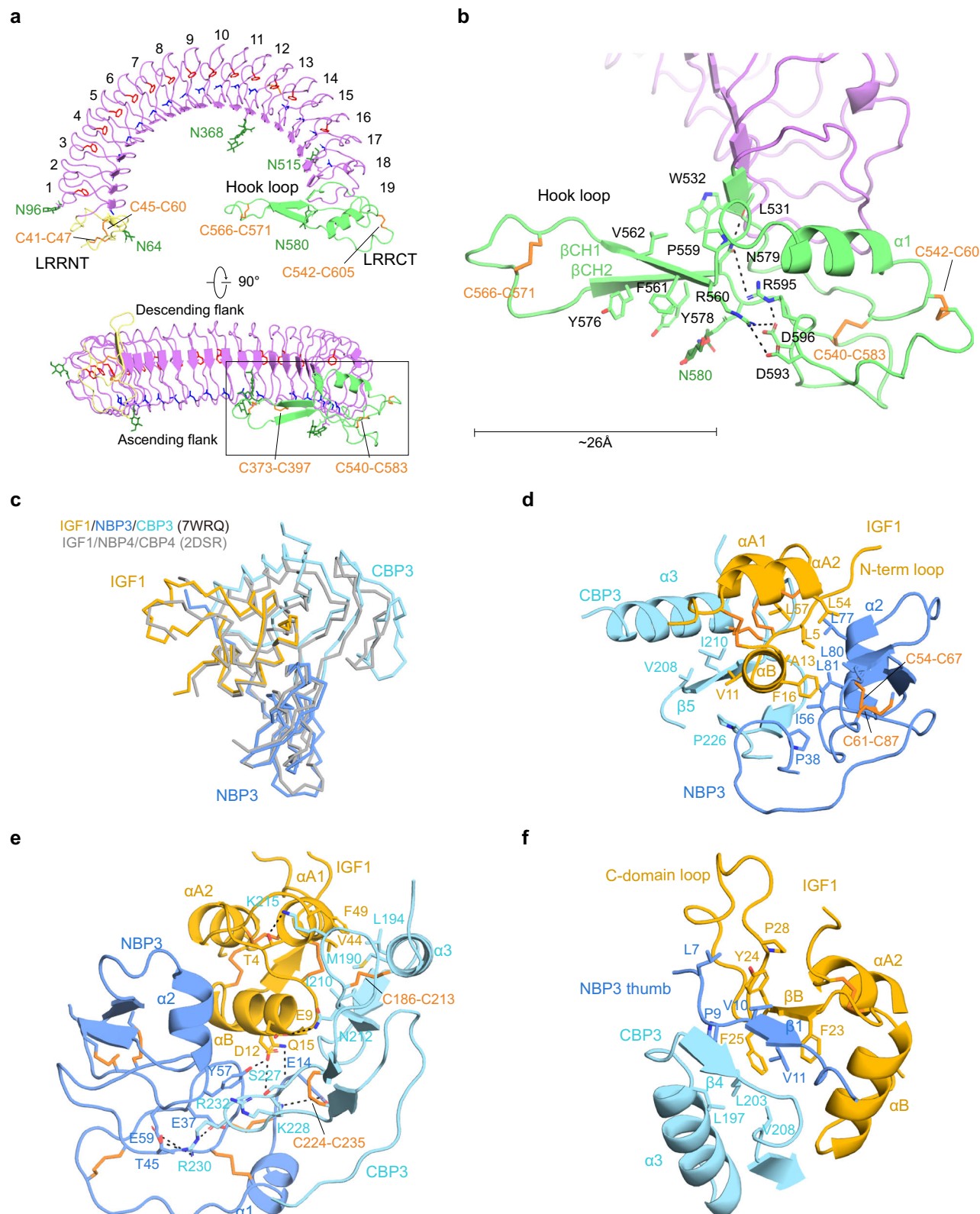

K228 and R230 on the tip of the hairpin-like loop formed additional charge interactions with NBP3 [K228 (CBP3)-E14 (NBP3); R230 (CBP3)-E37/T45/E59 (NBP3)] to further stabilize the NBP3–CBP3 interaction.

It has been reported that the extended N-terminal thumb of NBP is critical not only for regulation of IGF1 binding to IGF1R but also for interactions between NBP and CBP[12]. The thumb region consists of a short stretch of the very first N-terminal residues of IGFBPs that precede the first N-terminal cysteine (amino acids 1–12 in IGFBP3)[21].

Although most residues on the surface of the IGFBP3 cleft for IGF1 interaction are predominantly conserved in IGFBPs, the thumb regions of the different NBPs are of different amino acid (aa) lengths (NBP1, 4 aa; NBP2, 6 aa; NBP3, 12 aa; NBP4, 5 aa; NBP5, 6 aa; NBP6, 1 aa) (Supplementary Fig. 3b). Substitution of the IGFBP4 thumb with a corresponding region from IGFBP3 slightly enhanced the binding affinity to IGF1[12], suggesting that the sequence of the N-terminal thumb may confer unique properties on each IGFBP. Of note, our cryo-EM

**Fig. 2 | Structure of ALS and key determinants of IGF1/IGFBP3 binary complex.** **a** Two different views of the horseshoe-like structure of ALS. LRRNT (yellow), LRR 1–19 (violet), and LRRCT (lime) of ALS are labeled. The conserved asparagine-ladder (blue) and phenylalanine-spine (red) are shown as sticks. Descending and ascending flanks of ALS are labeled (bottom). **b** Close-up view of the ALS hook loop (boxed in A). Residues involved in stabilization of the ALS hook loop are displayed as sticks and labeled. **a**, **b** The glycosylation sites (N64, N96, N368, N515, and N580) and disulfide bridges (C41–C47 and C45–C60 in LRRNT, C373–C397 in LRR motif, C540–C583, C542–C605, and C566–C571 in LRRCT) are indicated. Disulfide bridges (orange) and N-linked glycans (green) are shown as sticks. **c** Comparison of the

IGF1/IGFBP3 in the ternary complex with the structure of IGF1/NBP4/CBP4 complex (gray, PDB: 2DSR). Structures are shown as ribbon diagrams. **d–f** Close-up views of the interacting interface between the IGF1 B domain α helix and the cleft formed by NBP3 and CBP3 (**d**), interaction at the interface between IGF1 and the entrance of the cleft, and direct contact between NBP3 and CBP3 (**e**), and β sheet formed by IGF1, CBP3, and the N-terminal thumb of NBP3 (**f**). The view for **e** is a rotation of the view for **d** -150° about the y-axis. Key residues involved in each interaction are displayed as sticks and labeled. The α helix (A8-V17) and β strand (G22-Y24) of the IGF1 B domain and the α1 helix (I43-F49) and α2 helix (L54-Y60) of the IGF1 A domain are labeled as αB, βB, αA1, and αA2, respectively.

structure identified a unique feature of the IGF1/IGFBP3 binary complex in which the NBP3 thumb is sandwiched between CBP3 β4 (N201-L203) and IGF1 B domain β strand (G22-Y24), forming a well-organized short β-sheet (Fig. 2f, Supplementary Figs. 3 and 4b). In particular, the NBP3 thumb (V10-V11) formed parallel and antiparallel β-sheets with CBP3 (N201-L203) and IGF1 (G22-Y24), respectively, whereas the NBP4 thumb formed a short parallel β-sheet only with CBP4. Moreover, this short β-sheet shielded the hydrophobic core consisting of NBP3 (P9 and V11), CBP3 (L197, L203, and V208), and IGF1 (F23 and F25). Another hydrophobic network [L7 and V10 (NBP3); Y24 and P28 (IGF1)] above this β-sheet would likely contribute to relocation of the IGF1 C-domain loop toward a more suitable position for interaction with ALS (Fig. 2f). In the following sections, the detailed interactions between the IGF1 C-domain loop and ALS are described. Given the conserved three hydrophobic residues at positions 2, 3, and 6 with respect to the first N-terminal Cys residue (L7, V10, and V11 for IGFBP3; L1, F4, and V5 for IGFBP5) (Supplementary Fig. 3b), we speculate that the NBP5 thumb would similarly be involved in the relocation of the IGF1 C-domain loop toward a more suitable position for interaction with ALS.

## The C-domain loop of IGF1 and CBP of IGFBP3 interact with ALS

Both IGF1 and IGFBP3 were bound to ALS with a buried surface area of 593 Å$^2$ and 1396 Å$^2$, respectively (Fig. 3a), and the IGF1 C-domain loop, a critical segment for recognizing IGF receptor, was held by ALS in the ternary complex (Fig. 3d). The interface between ALS and the IGF1/IGFBP3 binary complex could be divided into four distinct regions: IGF1 patch, CBP3 helix patch, CBP3 loop patch, and hook loop patch, all evolutionarily conserved (Fig. 3a, b and Supplementary Tables 2–4). Most of the interacting interfaces, except the CBP3 loop patch, were mediated by electrostatic interactions (Fig. 3c).

The first IGF1 patch was mediated by elaborate electrostatic networks involving residues of the IGF1 C domain (T29, R36, and R37) and ALS residues on descending flanks of LRR3–LRR6 (H126, S171, D174, and E198) (Fig. 3d and Supplementary Fig. 4d). W173 on ALS was wedged between R36 and R37 on IGF1, forming a hydrophobic interaction (Pi-Alkyl). These interactions allocated two arginines (R36 and R37) in opposite directions, enabling another electrostatic interaction between IGF1 and ALS [R37 (IGF1)-S171 (ALS); R36 (IGF1)-D174/E198 (ALS)]. Of note, Y31, R36, and R37 of IGF1, which are suggested to be involved in IGF1R binding[22,23], were identified as the key residues at the interface between IGF1 and ALS, indicating that ALS limits IGF binding to its receptor.

The second and third regions were formed by long-range interactions of CBP3 with the concave surface of ALS LRR5-LRRCT. In particular, the α3 helix of CBP3, forming the edge of the flat CBP3 segment, traversed from the ascending flank of LRR5 to the descending flank of LRR12 with a ~45-degree tilt toward the LRR β-strand. Extensive hydrogen bonds and ionic interactions of ALS LRR5–LRR12 with CBP3 α3 and its flanking loops were observed in the CBP3 helix patch (Fig. 3e and Supplementary Fig. 4e). The CBP3 loop patch was composed of contacts between ALS LRR13-LRRCT and the CBP3 C-terminal loop following the β7 strand (P244-K264). The LRR13–LRR14 and LRR19-LRRCT of ALS flanked both sides of the CBP3 loop through electrostatic interactions, stabilizing the weak central hydrophobic network

[Y462, W486, and Y510 (ALS LRR17–19); L245, P246, and Y248 (CBP3 loop)] (Fig. 3f and Supplementary Fig. 4e). Similar to other IGFBPs, the highly conserved disulfide bond (C237-C258) bridged the C-terminal end of CBP3 to its β7 (Fig. 3f and Supplementary Fig. 5).

Lastly, the hook loop and long glycan chain attached to N368 of ALS interacted with the IGF1/IGFBP3 binary complex. The hook loop of ALS was intercalated between NBP3 and CBP3 to tighten the ALS and IGFBP3 interactions. Although we cannot model the residues of the ALS hook loop tip unambiguously due to the lack of side chain densities, the backbone position of ALS hook loop can be determined by clear cryo-EM density (Supplementary Fig. 2d). Therefore, it is likely that the negative charged residues (E567 and D569) at the ALS hook loop tip contribute to the interaction with IGFBP3's positive charged residues [R19 (NBP3), R206 and R225 (CBP3)] (Fig. 3g and Supplementary Fig. 4e). It has been reported that complete de-glycosylation on ALS abolishes its ability to associate with IGFBP3[19]. Of interest, the long glycan chain at ALS N368 extended deep into the central axis of the ALS curvature where IGF1/IGFBP3 was located, and the first NAG formed a hydrogen bond with Q243 of CBP3 (Fig. 3f, g). Other than that, we observed no direct interaction between glycan chains of ALS and IGFBP3. Given that glycans, which are invisible in our cryo-EM map, can be further extended toward IGFBP3, the glycan chains on N368 have the potential to interact with IGFBP3, particularly CBP3. Collectively, most of the critical interaction residues on CBP3 are highly conserved in IGFBP3/5 (R188, E191, K198, and R206), but not in IGFBP1/2/4/6, providing a structural basis for the ternary complex that is mainly formed through these sequence-specific interactions between ALS and the CBP of IGFBP3/5 (Supplementary Fig. 3).

To investigate the significance of these interacting interfaces for ternary complex formation [ALS LRR3–6 motif/IGF1 C-domain loop (IGF1 patch), ALS LRR5–12 motif/CBP3 α3 helix (CBP3 helix patch), ALS LRR13-LRRCT and CBP3 C-terminal loop (CBP3 loop patch) and ALS hook loop/CBP3 loops/NBP3 α1 helix (hook loop patch)], we selected key interacting residues and introduced mutations (ALS W173A and IGF1 R36E/R37E for the IGF1 patch; IGFBP3 R188E for the CBP3 helix patch; ALS R414A for the CBP3 loop patch; ALS Δ561–578 deletion for the hook loop patch) (Supplementary Fig. 6). Then, we co-expressed the different combinations of IGF1-Strep-tag, IGFBP3, and ALS-His as indicated in Supplementary Fig. 6b (table) using Expi 293-F cells and monitored the components in the resulting complex by pull-down of IGF1 with Strep-Tactin resin. Although all mutants were well expressed and the binary complex (IGF1 R36E/R37E-IGFBP3 WT or IGF1 WT-IGFBP3 R188E) was properly assembled, the amount of ALS bound to these binary complexes were significantly reduced (Supplementary Fig. 6b), implying that these residues in each interacting patch are indeed critical for ternary complex formation. Particularly, mutations in the CBP3 helix patch and CBP3 loop patch (IGFBP3 R188E for the CBP3 helix patch, and ALS R414A for the CBP3 loop patch) completely abolished ALS binding to the binary complex, indicating that interactions between CBP3 and the ALS concave surface are especially crucial for ternary complex formation and partially explaining why IGF1 alone could not bind to ALS. Collectively, these results suggest that the IGF1/IGFBP3 binary complex mainly binds to the ALS concave surface, with

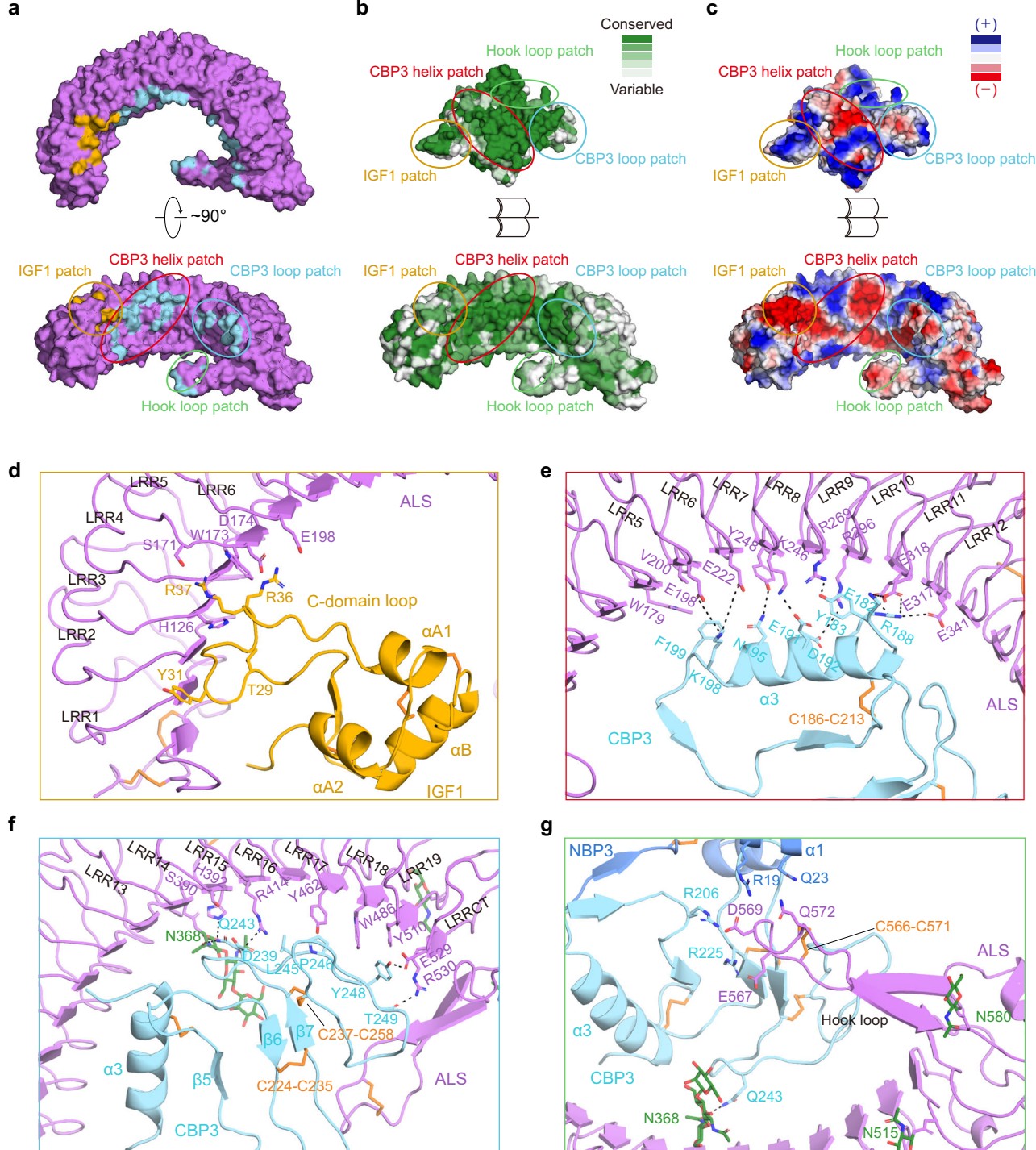

**Fig. 3 | Detail of interactions between ALS and the IGF1/IGFBP3 binary complex. a** The surface representation of ALS (violet) in two orthogonal views. ALS residues for IGF1 and CBP3 interaction are shown in yellow and cyan, respectively. **b** Sequence conservations were calculated by the Consurf server[65] and are presented on the surfaces of ALS and the IGF1/IGFBP3 binary complex. The structures are shown as open-book views and surface representations, with color maps reflecting conservation (green, highly conserved; white, less conserved). The orientation of the surface view of ALS is identical to that in **a** (bottom). **c** Electrostatic potential of ALS and the IGF1/IGFBP3 binary complex, calculated according to the Poisson-Boltzmann equation in PyMOL. Blue and red represent positively and negatively charged residues, respectively. The orientation of the

surface view of ALS and the IGF1/IGFBP3 binary complex are identical to that in **b**. **a**–**c** The IGF1, CBP3 helix, CBP3 loop, and hook loop patches are marked with circles colored yellow, red, cyan, and lime, respectively. **d**–**g** Close-up views of interactions between ALS and the IGF1/IGFBP3 binary complex. Interactions (**d**) between the ALS LRR3–6 motif and IGF1 C-domain loop (IGF1 patch), (**e**) between the concave surface of the ALS LRR5–12 motif and CBP3 α3 helix (CBP3 helix patch), (**f**) between the concave surface of the ALS LRR13-LRRCT and CBP3 C-terminal loop (CBP3 loop patch), and (**g**) interactions among the ALS hook loop, CBP3 loops, and NBP3 α1 helix (hook loop patch). Key residues involved in each interaction are displayed as sticks and labeled.

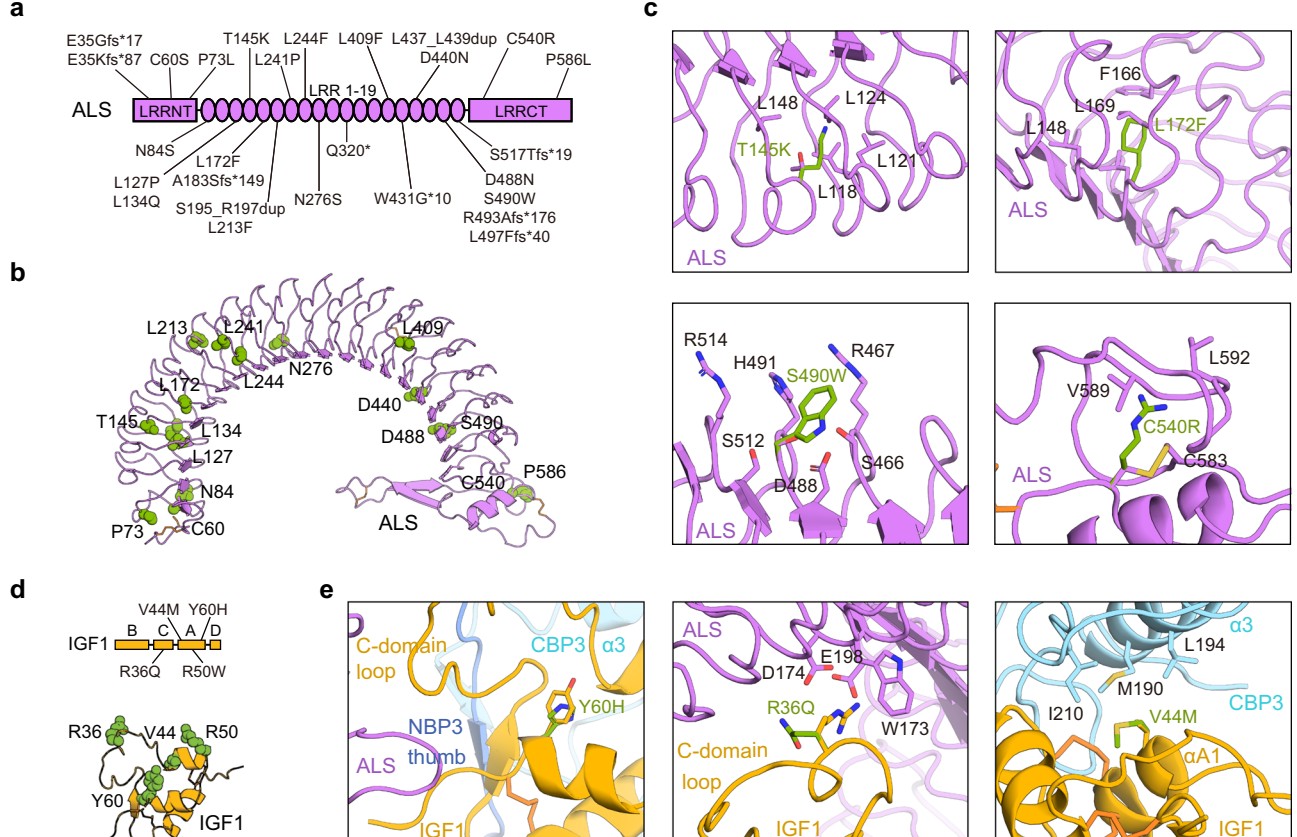

**Fig. 4 | Structural basis for disease-associated ALS and IGF1 mutations.**
**a** Schematic representation of the ALS protein indicating the location of the 27 identified mutations from patients with ASCLD. **b** Mutated residues in patients with ASCLD (only point mutations) are presented as green spheres and labeled in the cartoon structure of human ALS. **c** Close-up views of four cases (T145K, L172F, S490W, and C540R) of ALS mutations and their neighboring residues. **d** Mutated IGF1 residues in human patients with growth impairment are indicated in a schematic representation (left) and presented as green spheres and labeled in a cartoon structure of human IGF1 (right). **e** Close-up views of three IGF1 mutants (Y60H, R36Q, and V44M) and their neighboring residues.

further stabilization by interactions with the hook loop on LRRCT and the glycan attached to N368 of ALS.

**Structural basis for disease-associated ALS and IGF1 mutations**
To date, various mutations (17 missense, 7 frameshift, 2 in-frame insertion, 1 nonsense mutation, and 1 deletion of exon 2) in the human *IGFALS* gene have been reported in patients with ACLSD[16,24] (Fig. 4a, b). Although these ALS missense mutations are located in the LRRNT (C60S and P73L), LRR (N84S, L127P, L134Q, T145K, L172F, L213F, L241P, L244F, N276S, L409F, D440N, D488N, and S490W), and LRRCT domains (C540R and P586L), none of residues are involved in direct interactions with the IGF1/IGFBP3 binary complex. Our cryo-EM structure clearly showed that these mutations can affect the structural integrity of ALS, possibly causing its misfolding and aberrant secretion (Fig. 4c), which are consistent with a previous analysis using the predicted ALS structural model[20]. In particular, the mutation on Leu and Asn residues (N84S, L127P, L134Q, L172F, L241P, L244F, N276S, and L409F) as part of the LRR consensus sequence (xLxxLxLxxNxLxxLxxxxFxx/Lx) are likely to introduce a main chain distortion or disrupt the LRR hydrophobic core, affecting the entire LRR architecture of the ALS protein. Similar to this effect, substitution of T145 and L213 residues to long charged Lys and bulky hydrophobic Phe, respectively, which point toward the hydrophobic core of the LRR solenoid interior, would introduce steric hindrance with surrounding residues and destroy the regular fold of the LRR module. The highly conserved C60 and C540 residues participate in disulfide bridge formation with C45 and C583, respectively, and the P73 and P586 residues contribute to maintaining structural rigidity. Therefore, mutations at

these residues (C60S, C540R, P73L, and P586L) could destabilize LRRNT and LRRCT (Fig. 4c). Indeed, previous studies on the in vitro expression of ALS variants demonstrated that N276S, L409F, and C540R are not synthesized, whereas L213F is partially expressed but cannot be secreted[24]. Although D440 and D448 of ALS are not essential to ALS/IGFBP3 interactions, D440N and possibly D488N could introduce a new N-glycosylation site that leads to impairment of secretion of ALS and its ability to form a ternary complex, as reported previously[25]. The S490W mutation is predicted to cause loss of the hydrogen bond network with neighboring S466, R467, D488, H491, S512, and R514 on the surface of the LRR17–LRR19 ascending flank (Fig. 4c). Two in-frame mutants, S195_R197dup and L437_L439dup, insert three amino acids at LRR6 and LRR16, respectively. The resulting altered length of the LRR modules might impair alignment of the hydrophobic residues on the consensus motif, perturbing the solenoid-like structure of ALS. Moreover, S195_R197dup and L437_L439dup lie adjacent to the ternary complex interface, probably disrupting ternary complex formation.

IGF1 deficiency and several bio-inactive IGF1 mutations (R36Q, V44M, R50W, and Y60H) also have been identified in patients with growth failure[26–28] (Fig. 4d). IGF1 Y60, which is highly conserved among species and also conserved in IGF2 and insulin, is far away from the interaction interfaces of the ternary complex, whereas IGF1 R36 and V44 lie near the IGF1 interaction interface with ALS and CBP3, respectively. However, none of these mutations (R36Q, V44M, R50W, and Y60H) seem to affect IGF1 binding to ALS or IGFBP3[29,30] (Fig. 4e). Therefore, their decreased affinity to IGF1R may result in phenotypes such as severe intrauterine and postnatal growth retardation,

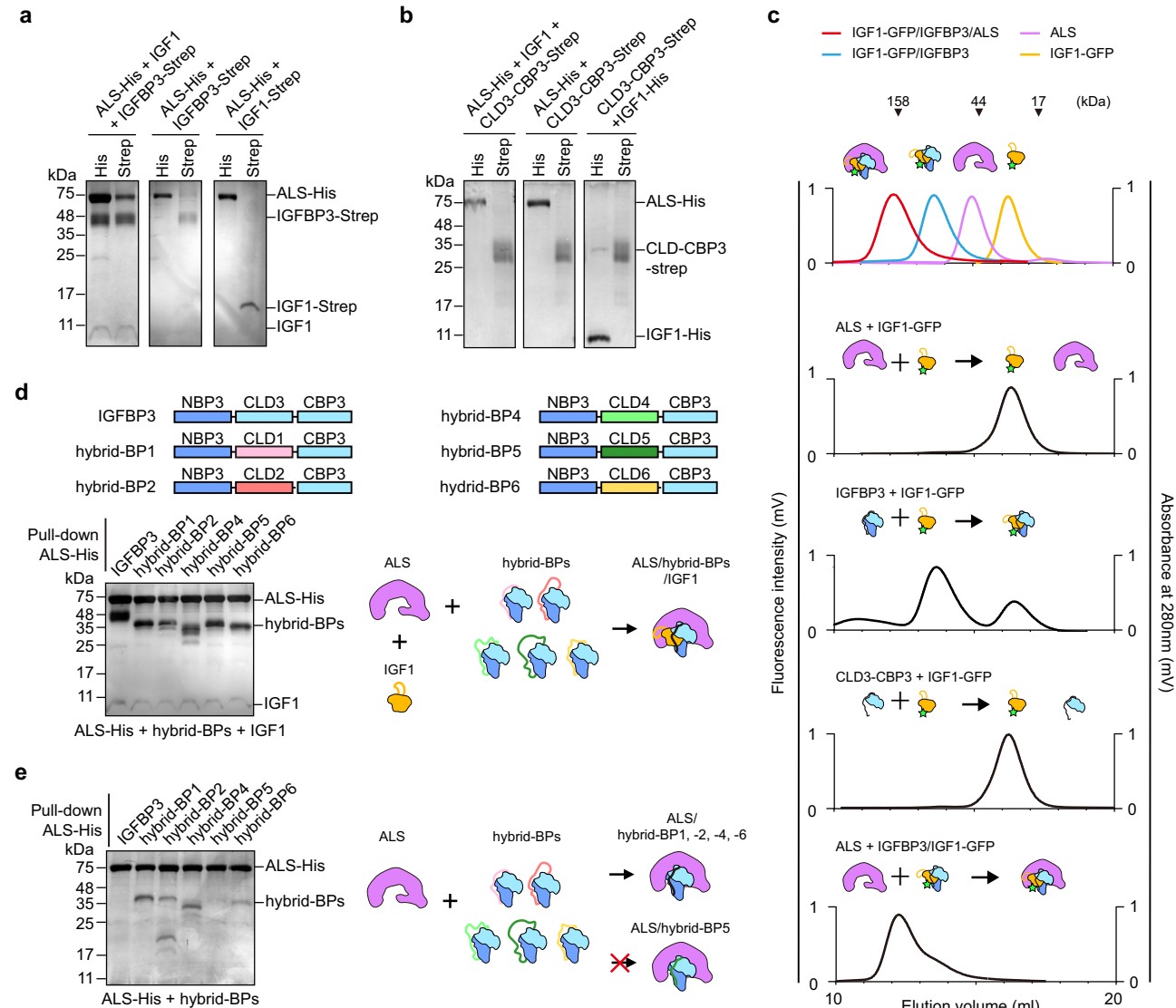

**Fig. 5 | Critical role of CLD3 for sequential assembly of IGF1/IGFBP3/ALS ternary complex. a** SDS-PAGE analysis of the co-expression test. ALS-His, IGFBP3-Strep, and IGF1 (IGF1 or IGF1-Strep) are co-expressed and pulled down with Ni-NTA or Strep-Tactin resin, as indicated. **b** Co-expression of CLD3-CBP3 with ALS and/or IGF1. ALS-His, CLD3-CBP3-Strep, and IGF1 (IGF1 or IGF1-His) were co-expressed and pulled down with Ni-NTA or Strep-Tactin resin, as indicated. **c** Fluorescence-detection size-exclusion chromatography (FSEC) after in vitro reconstitution of indicated proteins. Fluorescence signal (GFP) fused to IGF1 was monitored to examine the complex formation. The elution profiles for standard IGF1-GFP

(yellow), IGF1-GFP/IGFBP3 binary complex (blue), IGF1-GFP/IGFBP3/ALS ternary complex (red), and ALS alone (purple) are indicated as a control (top). **d** Co-expression of hybrid-BPs with ALS and IGF1. Schematic diagram of the human IGFBP3 and hybrid-BPs (top). Each domain is labeled above the diagram. ALS-His and IGF1 were co-expressed with IGFBP3 (lane 1) or hybrid-BPs (lanes 2–6) and pulled with Ni-NTA resin, then analyzed by SDS-PAGE. **e** Co-expression of hybrid-BPs with ALS. ALS-His were co-expressed with IGFBP3 (lane 1) or hybrid-BPs (lanes 2–6) and pulled with Ni-NTA resin, then analyzed by SDS-PAGE. **a**, **b**, **d**, **e** Similar results were observed in three independent experiments.

microcephaly, and sensorineural deafness[22,26]. Taken together, our findings provide a structural basis for ACLSD and the impaired growth seen with genetic defects in the human IGF1/IGFBP3/ALS ternary complex.

## The CLD of IGFBP3 is crucial for sequential assembly of IGF1 and IGFBP3 with ALS

Although the high-resolution cryo-EM structure of IGF1/IGFBP3/ALS elucidated interactions among the components of the ternary complex, the structure itself is insufficient to explain the assembly process for the IGF1/IGFBP3/ALS ternary complex. To address this gap, we co-expressed the different combinations of IGF1, IGFBP3, and ALS in Expi 293-F cells and monitored the components in the resulting complex (Fig. 5a). As His-tag and Strep-tag are attached to the C-terminus of ALS and IGFBP3, respectively, we could pull down ALS or IGFBP3 as well as

the complexes containing ALS or IGFBP3 using Ni-NTA or Strep-Tactin resin. When three proteins were co-expressed, IGF1, IGFBP3, and ALS bands were all observed in both His and Strep pull-down, indicating proper assembly of the ternary complex. As IGFBP3 engages in long-range interactions with the concave surface and hook loop of ALS in the ternary complex (Fig. 3), we speculated that IGFBP3 in the absence of IGF1 could bind to ALS. However, when ALS and IGFBP3 were co-expressed without IGF1, only ALS in His pull-down or IGFBP3 in Strep pull-down was detected, indicating that ALS and IGFBP3 cannot form a binary complex without IGF1. The results of co-expression of ALS and IGF1 also demonstrated that IGF1 alone cannot form a complex with ALS, consistent with previous reports[14]. Moreover, we found that CLD3-CBP3 cannot bind to both ALS and IGF1 (Fig. 5b), suggesting that NBP3 is required for IGF1/IGFBP3 binary complex formation. Unfortunately, we could not test whether CBP3 alone is sufficient for ALS

binding, because we were unable to express CBP3 probably due to the exposed hydrophobic residues on its flat surface.

To assess the quantitative efficiency of IGF complex assembly, we performed fluorescence-detection SEC (FSEC) after in vitro reconstitution with different combinations of recombinant proteins (ALS, IGFBP3, CLD-CBP3, IGF1-GFP, and IGF1-GFP/IGFBP3 binary complex) and monitored fluorescence signal from the green fluorescent protein (GFP) fused to IGF1. Similar to co-expression and pull-down analysis, we found that IGF1-GFP show no affinity for ALS and CLD-CBP3, and that assembly of the IGF1/IGFBP3 binary complex should precede formation of the IGF1/IGFBP3/ALS ternary complex (Fig. 5c).

The CLD sequences of six members of IGFBP are less conserved compared with their NBPs and CBPs (Supplementary Fig. 3b). Therefore, we next questioned whether CLD could influence the specific binding of the IGF1/IGFBP3 binary complex to ALS. To this end, we replaced the CLD of IGFBP3 with that of a different IGFBP and used co-expression and pull-down assays to examine whether these hybrid-BPs could form a ternary complex with IGF1 and ALS. Similar to IGFBP3, all versions of the hybrid-BP could form the ternary complex with IGF1 and ALS (Fig. 5d), indicating that the CLD does not confer specificity of each IGFBP for ternary complex formation with ALS. Surprisingly, we found that even in the absence of IGF1, ALS could bind to hybrid-BP1, -2, -4, and -6, but not to IGFBP3 or hybrid-BP5 (Fig. 5e), suggesting that the CLDs of IGFBP3 and IGFBP5 sterically block the interface for ALS binding. Collectively, our results imply that IGF1 binding to the cleft between NBP3 and CBP3 may induce conformational changes in CLD3, facilitating ALS binding to the pre-assembled IGF1/IGFBP3 binary complex for formation of the IGF1/IGFBP3/ALS ternary complex.

## IGFBP3 CLD proteolysis by thrombin and ADAM12 induces IGFBP3 CBP release and produces the intermediate ternary complex

Proteolysis at the IGFBP CLD by proteases such as metalloproteinase, cathepsins, and serine proteases is crucial for activation of IGFR signaling in a target tissue or cell[10,31,32]. To check whether IGFBP proteolysis causes IGF1 release from the ternary complex, we incubated the purified IGF1-GFP/IGFBP3 binary complex or IGF1-GFP/IGFBP3/ALS ternary complex with thrombin, which cleaves R97-A98 and R206-G207 of IGFBP3[32], and used FSEC to monitor IGF1-GFP release from the complex. As expected, IGF1-GFP was released from the binary complex after thrombin digestion and eluted at the peak position, corresponding to free IGF1-GFP (Fig. 6a, middle). Strikingly, IGF1-GFP peak released from the ternary complex after thrombin digestion was barely detected (Fig. 6a, bottom and Supplementary Fig. 7a), in contrast to previous reports[33]. Instead, the fluorescence peak was slightly shifted to the smaller complex (~140 kDa) (Fig. 6a), suggesting that IGF1 is not released from the ternary complex after IGFBP3 proteolysis, but that some other components of the ternary complex might be released. To identify the released component, we performed immunoblot analysis on the SEC fraction of the IGF1/myc-IGFBP3-strep/ALS ternary complex after cleavage with thrombin, using antibodies against ALS and IGF1 as well as anti-myc and anti-Strep antibody for NBP3 and CBP3 detection, respectively. Of interest, the CBP3 band was not detected after thrombin digestion and subsequent SEC, whereas ALS, IGF1, and cleaved NBP3 remained in the ternary complex (Fig. 6b). These results indicate that CBP3, not IGF1, is released from the ternary complex after IGFBP3 proteolysis. FSEC analysis with IGF1/IGFBP3-GFP/ALS demonstrated that the amount of released CBP3-GFP from the ternary complex upon thrombin proteolysis was increased in time-dependent manner (Fig. 6c).

It has been known that a disintegrin and metalloprotease 12 (ADAM12) and pregnancy-associated plasma protein A2 (PAPP-A2) in pregnancy serum are also responsible for proteolysis of IGFBP3 and IGFBP5[34,35], and mutations in PAPP-A2 lead to short stature because of low IGF1 availability (OMIM# 619489)[36]. Therefore, we further analyzed

IGFBP3 proteolysis by ADAM12 and PAPP-A2, and subsequent CBP3 release from the ternary complex. Similar to thrombin, ADAM12 efficiently degraded IGFBP3 in the ternary complex, which in turn induced the release of CBP3-GFP, but not IGF-GFP (Supplementary Fig. 7b–d). However, compared with thrombin and ADAM12, the efficiency of IGFBP3 cleavage by PAPP-A2 was significantly reduced (Supplementary Fig. 7b), which are consistent with the previous reports[37]. Interestingly, even after PAPP-A2 treatment to the IGF1/IGFBP3/ALS ternary complex, neither CBP3-GFP nor IGF-GFP were released (Supplementary Fig. 7c, d).

We then questioned whether the resulting intermediate complex (IGF1/NBP3/ALS) after proteolysis can activate IGF1R signaling. To this end, we treated HEK293 cells with the SEC fraction containing this intermediate ternary complex upon proteolysis with thrombin, ADAM12, or PAPP-A2 and monitored phosphorylation of IGF1R. Indeed, the original ternary complex itself did not induce IGF1R phosphorylation, but the SEC fraction containing the intermediate ternary complex after proteolysis with thrombin and ADAM12 could activate IGF1R signaling at a level similar to that with free IGF1 (Fig. 6d and Supplementary Fig. 7e). Of note, the intermediate ternary complex after PAPP-A2 proteolysis, which retained CBP3, could activate IGF1R signaling, but with less potency as compared with those of the intermediate ternary complex lacking CPB3 after thrombin and ADAM12 proteolysis (Fig. 6d and Supplementary Fig. 7e). Interestingly, the intermediate ternary complex lacking CBP3 after thrombin proteolysis could not directly bind to the purified IGF1R ectodomain dimer (IGF1R ectodomain-leucine zipper)[38], whereas IGF1 alone could do so (Supplementary Fig. 7f). Moreover, the presence of the IGF1R ectodomain alone could not promote the quick dissociation of IGF1 from this intermediate ternary complex (Supplementary Fig. 7f). These results suggest that after proteolysis of IGFBP3 CLD and CBP3 release, IGF1 needs to be further dissociated from the destabilized IGF1/IGFBP3-NBP/ALS complex for IGF1R activation, which may be mediated by some other unknown factors at the extracellular matrix (ECM) or plasma membrane.

To elucidate the specific role of the IGFBP CLD on CBP release, we incubated ternary complexes containing hybrid-BPs, IGF1, and ALS with thrombin and treated HEK293 cells with them. Strikingly, although all CLDs in hybrid-BPs were properly cleaved by thrombin, only the ternary complex containing IGFBP3 or hybrid-BP5 strongly activate IGF1R upon thrombin digestion (Fig. 6e, f and Supplementary Fig. 7g). Consistent with FSEC results of IGF1-GFP/IGFBP3/ALS upon proteolysis, IGF1-GFP is not released from the IGF-GFP/hybrid-BPs/ALS ternary complex after CLD proteolysis with thrombin, ADAM12 and PAPP-A2 (Fig. 6g and Supplementary Fig. 7h–j). More interestingly, the amount of released CBP3-GFP from the ternary complex containing hybrid-BPs after thrombin digestion was highly correlated with their ability for IGF1R phosphorylation (hybrid-BP5 ≫ hybrid-BP2 and -4 > hybrid-BP1 and -6) (Fig. 6f, h), suggesting that the release of CBP3 from the ternary complex after CLD proteolysis destabilizes the intermediate complex and further enhances IGF1 bioavailability. Taken together, our findings indicate that the CLDs of IGFBP3 and IGFBP5 play a crucial role not only in sequential assembly of the ternary complex, but also in disassembly of CBP3 after CLD proteolysis for IGF bioavailability.

## Discussion

In this study, we determined the cryo-EM structure of the human IGF1/IGFBP3/ALS ternary complex. We show that the IGF1/IGFBP3 binary complex, in which IGF1 is clamped by the NBP and CBP of IGFBP3, binds to the concave surface of the horseshoe-like ALS, forming a parachute-like ternary complex. We also identified the important structural features of ALS such as the long hook loop on LRRCT and the glycans attached to N368, which have not been assessed by previous in silico structural model[20]. Moreover, our in vitro biochemical

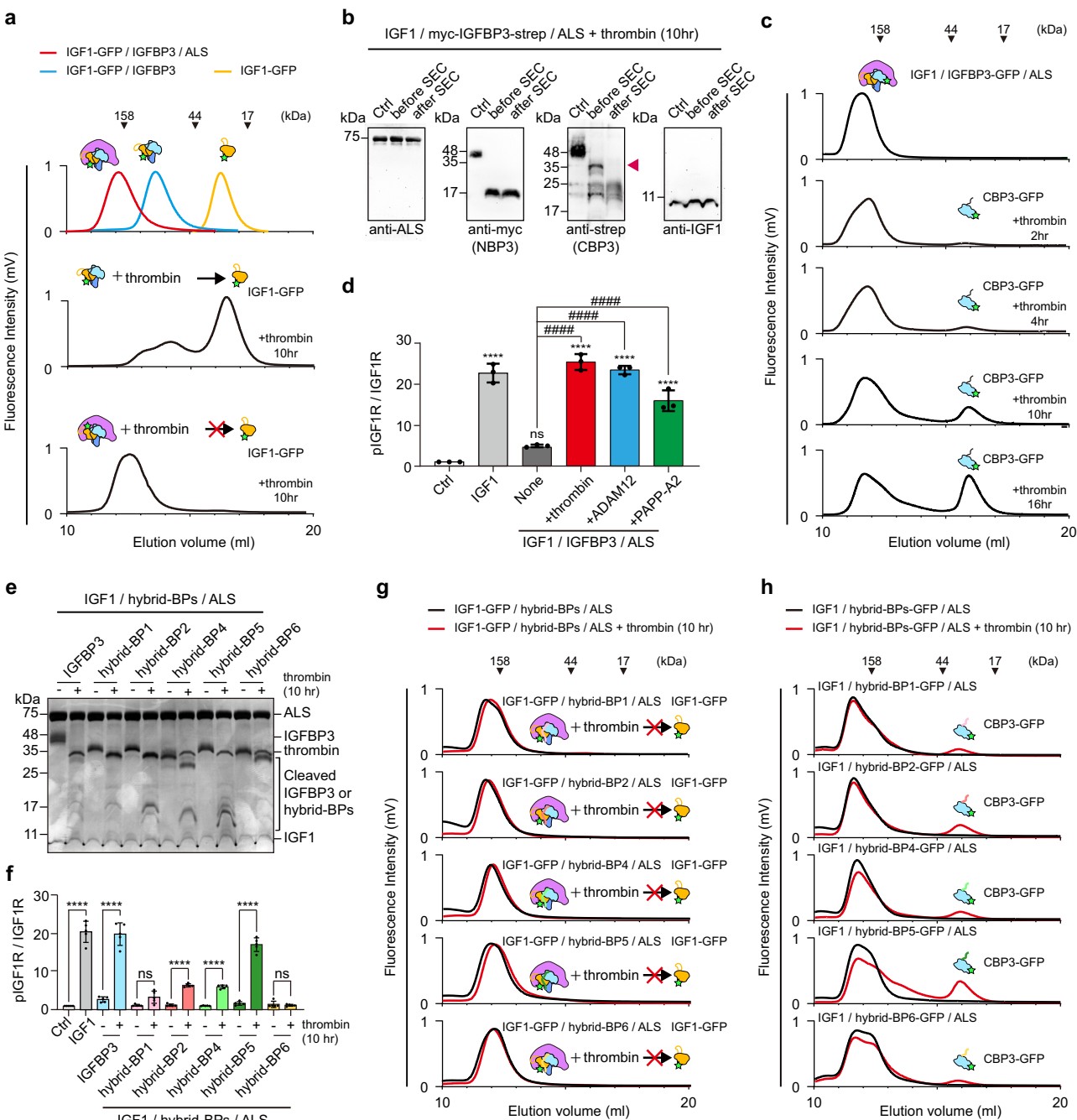

**Fig. 6 | Release of CBP3 from the ternary complex after CLD3 proteolysis.**
**a** Fluorescence-detection size-exclusion chromatography (FSEC) of binary or ternary complexes after 10 h proteolysis by thrombin. The elution profile for standard IGF1-GFP (yellow), IGF1-GFP-strep/IGFBP3 binary complex (blue), and IGF1-GFP/IGFBP3/ALS-His ternary complex (red) are indicated as a control (top). **b** Immunoblot analysis of IGF1/myc-IGFBP3-Strep/ALS complex after 10 h proteolysis by thrombin and subsequent SEC. Anti-myc and anti-Strep antibody were used to detect NBP3 and CBP3, respectively. No-thrombin treatment served as negative control (Ctrl). **c** FSEC of the ternary complex (IGF1/IGFBP3-GFP/ALS-His) after thrombin digestion for indicated time. Fluorescence signal (GFP) was monitored to examine the dissociation of CBP3 from the ternary complex. **d** Quantification of immunoblot analysis for IGF1R phosphorylation after treatment of intermediate ternary complex. HEK293A cells were treated with IGF1-His (5 nM, positive control), IGF1-His/IGFBP3/ALS ternary complex (5 nM, none), or intermediate ternary complex (5 nM). The intermediate ternary complex was prepared by protease digestion (thrombin, ADAM12 and PAPP-A2) and subsequent SEC purification. No treatment was used for negative control (Ctrl). Data from three independent experiments

($n = 3$) were analyzed and relative pIGF1R/IGF1R (Fold to Ctrl) were expressed as mean ± SD (ns, not significant vs. ctrl; ****$P < 0.0001$ vs. ctrl; ####$P < 0.0001$ vs. none). *P*-values by one-way ANOVA test followed by Sidak's multiple comparisons test. **e** SDS-PAGE analysis of the IGF1/hybrid-BPs/ALS ternary complex after thrombin digestion. **f** Quantification of immunoblot analysis for IGF1R phosphorylation with intermediate ternary complex containing hybrid-BPs. IGF1-His (5 nM, positive control), IGF1-His/IGFBP3/ALS ternary complexes (5 nM), or IGF1-His/hybrid-BPs/ALS ternary complexes (5 nM) with and without thrombin digestion were treated onto HEK293A cells. No treatment for negative control (Ctrl). Data from five independent experiments ($n = 5$) were analyzed and relative pIGF1R/IGF1R (Fold to Ctrl) were expressed as mean ± SD (ns, not significant; ****$P < 0.0001$ Ctrl vs. IGF1-His or -thrombin vs. +thrombin for each group). *P*-values by one-way ANOVA test followed by Sidak's multiple comparisons test. **g, h** FSEC of the ternary complex IGF1-GFP/hybrid-BPs/ALS-His (**g**), or IGF1/hybrid-BPs-GFP/ALS-His (**h**) after thrombin digestion. Fluorescence signal (GFP) was monitored to examine the dissociation of IGF1 or CBP3 from the indicated ternary complex.

experiments with hybrid-BPs showed that CLD in IGFBP3 is crucial for sequential assembly of the ternary complex and CBP3 release from the complex after proteolysis for IGF1R activation.

The IGF1/IGFBP binary complex, which adopts similar global folds with the IGF1/IGFBP4 (NBP4/CBP4) complex, binds to the concave surface of ALS through the IGF1 C-domain loop and the IGFBP α3 helix and C-terminal loop. A 3D structural comparison using the DALI server[39] indicated that the overall structure of ALS is similar to that of other LRR superfamily proteins such as drosophila Toll (PDB ID: 4LXR; Z-score: 29.3; RMSD: 4.9), LINGO1 (PDB ID: 4OQT; Z-score: 28.2; RMSD: 4.9), and LRIM1 (PDB ID: 3J0A; Z-score: 24.0; RMSD: 3.5), but ALS structure is unique in that it has a long hook loop on LRRCT protruding into the central axis of the LRR curvature. Although this hook loop does not directly interact with IGF1, it further stabilizes the interaction between ALS and the binary complex by intercalating between NBP3 and CBP3. Our ternary complex structure clearly shows that the hydrophobic patch of the IGF1 B domain as well as Y31, R36, and R37 of the IGF1 C domain, which are crucial for IGF1R binding[13,40], are buried, indicating that the ternary complex sterically protects IGF1 from IGF1R binding. The residues on IGF1 for IGFBP3 interactions are highly conserved in IGF2, suggesting that the IGF2/IGFBP3 binary complex would adopt a conformation comparable to that of IGF1/IGFBP3. However, the C domain of IGF2 is shorter than that of IGF1, and only two of four residues needed for interaction between ALS and IGF1 are conserved in IGF2, explaining its weaker affinity compared with IGF1 for ALS/IGFBP3[41].

We also provide a structural basis for the role of N-linked glycan attached to ALS in ternary complex formation. Our structural analysis of the human IGF1/IGFBP3/ALS ternary complex identified five N-linked glycosylation sites on ALS (N64, N96, N368, N515, and N580) among seven potential candidates. Janosi et al. have shown that individual mutations of these N-linked glycosylation sites do not have a major effect on ALS binding to the IGF1/IGFBP3 complex, while enzymatic removal of these glycans abolishes complex formation[19]. These findings indicate that a number of N-linked glycans of ALS rather than any single specific carbohydrate chain are responsible for its ability to form a ternary complex. Indeed, N-acetylglucosamine (NAG) attached to ALS N580 contribute to stabilizing the ALS hook loop, allowing it to interact with IGFBP3. Moreover, although only one mannose and two NAGs attached to ALS N368 are visible in cryo-EM density, further extended glycans from the terminal mannose, such as galactose and sialic acid, are likely to interact with positively charged residues of IGFBP3, such as H196 and R206, supporting previous findings that removal of sialic acids from ALS decreases its affinity for complex formation by 50−80%[19]. On the other hand, we could not observe N-linked glycans on IGFBP3 because three glycosylation sites (N89, N109, and N172) are located on the flexible IGFBP3 CLD and are invisible in cryo-EM density. These results support previous findings that N-linked glycans on IGFBP3 are non-essential to ALS or IGF binding, but they may modulate other biological activities of IGFBP3 such as ECM binding[42].

Among six members of the IGFBP family, only IGFBP3 and IGFBP5 can form a ternary complex with IGF and ALS[14]. Our cryo-EM structure of the human IGF1/IGFBP3/ALS ternary complex provides a clear understanding of what would confer this unique ability on IGFBP3 and IGFBP5. First, the hydrophobic long N-terminal thumb in IGFBP3 (also in IGFBP5) provides local steric constraints on IGF1 C-domain loop, thus facilitating relocation and anchoring of the IGF1 C domain toward ALS in a suitable configuration (Figs. 2f and 7a). As the C-domain loops of IGF1 are partially visible in previous apo IGF1 structures due to its high flexibility[9,43], we were unable to compare entire C-domain loops in apo-IGF1 with those in the ternary complex. However, a structural comparison of the IGF1 C-domain loop in the ternary complex with that of the IGF1/NBP4 complex (PDB ID: 1WQJ)[21] indicates that the protruding C domain of IGF1 in IGF1/NBP4 complex is likely to trigger a steric crash, disrupting potential interactions with ALS, whereas the

IGF1 C-domain loops tightly contacts ALS in the IGF1/IGFBP3/ALS ternary complex (Fig. 7a). Second, the C-terminal loop of CBP3 following the β7 strand (P244-K264), which interacts with the ALS concave surface, is structurally flexible and extended (Figs. 3f and 7b). In contrast, the corresponding regions in other CBP/IGF1 complexes are rigid secondary structures [$3_{10}$ turn in CBP4 (PDB ID: 2DSR) and β-strand in CBP1 (PDB ID: 2DSQ)[12]], taking on a more compact conformation and hindering close contacts with the ALS concave surface (Fig. 7b and Supplementary Fig. 5). Previous domain swapping experiments of IGFBP3 with IGFBP2 or IGFBP6, which cannot form ternary complexes, have demonstrated that the CBP of IGFBP3 and IGFBP5 are crucial for ALS binding[44,45]. Indeed, the apo-structures of CBP2 and CBP6, particularly the structure of their C-terminal loop region, are quite different from that of CBP3 (Cα RMSD CBP3 vs. CPB2: -3.118 Å; CBP3 vs. CPB6: -3.789 Å). Lastly, the key residues of IGFBP3 constituting CBP3 helix patch interactions, such as R188, E191, and K198, are highly conserved in IGFBP5 but not in other IGFBPs. These unique features of IGFBP3 and IGFBP5 enable their specific binding to ALS and IGF, thus forming the ternary complex.

Previous studies showed that isolated NBP and CBP of IGFBPs have significantly lower affinities for IGFs than do intact IGFBPs[46,47], suggesting that the flexible CLD of IGFBPs plays a role in formation of the IGF/IGFBP binary complex, perhaps by controlling inter-domain movement for their cooperative binding to IGF. As all cases of hybrid-BP, in which CLDs of IGFBP3 are replaced by those of other IGFBPs, can form the ternary complex with IGF1 and ALS, a less conserved CLD could not be the reason for preferential binding of ALS only to IGFBP3 and IGFBP5. Instead, the current study implies a unique role for the IGFBP3 CLD in sequential assembly of the ternary complex. Our co-expression and pull-down analysis results demonstrate that ALS has a low affinity for uncomplexed IGFBP3 and hybrid-BP5, but a high affinity for uncomplexed hybrid-BPs (hybrid-BP1, -2, -4, and -6). This finding suggests that the CLDs of IGFBP3 and IGFBP5 sterically block the ALS-binding interface and that binary complex assembly may induce conformational changes in CLD3, thus exposing the CBP surface for ALS binding to form the ternary complex (Fig. 7c, top). These steps illustrate why the assembly of the binary complex (IGF1/IGFBP3 and IGF1/IGFBP5) should precede the formation of the ternary complex (IGF1/IGFBP3/ALS and IGF1/IGFBP5/ALS). However, we do not exclude the possibility that the unique features such as amino acid composition and length of the IGFBP3 CLD and/or the conformational changes in the CBP3 loop patch of IGFBP3 upon IGF1 binding are also involved in sequential assembly of the ternary complex.

It has long been known that IGFBP proteolysis of the central linker domain leads to increased concentrations of bioavailable IGF and subsequent activation of the IGF1R[48]. Surprisingly, we found the intermediate ternary complex after proteolysis by thrombin and ADAM12, in which IGF1 is retained, but IGFBP3 CBP3 is released, can stimulate IGF1R phosphorylation. These intermediate ternary complexes can be explained by the previous results showing that CBPs have lower affinities than NBPs for IGF1[46,47]. Moreover, with FSEC and IGF1R activation analysis using hybrid-BPs, we showed that CLD3 and CLD5 have unique properties for CBP dissociation from the ternary complex. Because of the different exclusive digestion sites, proteolysis with PAPP-A2 produced different fragments of IGFBP3 and the intermediate complex retaining CBP3. Moreover, the IGF1R activation potency of the PAPP-A2-treated ternary complex was less than those treated with thrombin and ADAM12, probably due to the limited proteolytic activity of PAPP-A2 against IGFBP3. However, our results suggest that proteolysis of the IGFBP3 CLD is required for IGF1R activation and that after the IGFBP3 CBP falls apart from the ternary complex by proteolysis (by thrombin or ADAM12), the remaining IGF1/IGFBP3-NBP/ALS complex will become relatively unstable, thereby further enhancing IGF1 bioavailability. We also postulate that the potential conformational changes of IGFBP3 CLD after proteolysis by thrombin

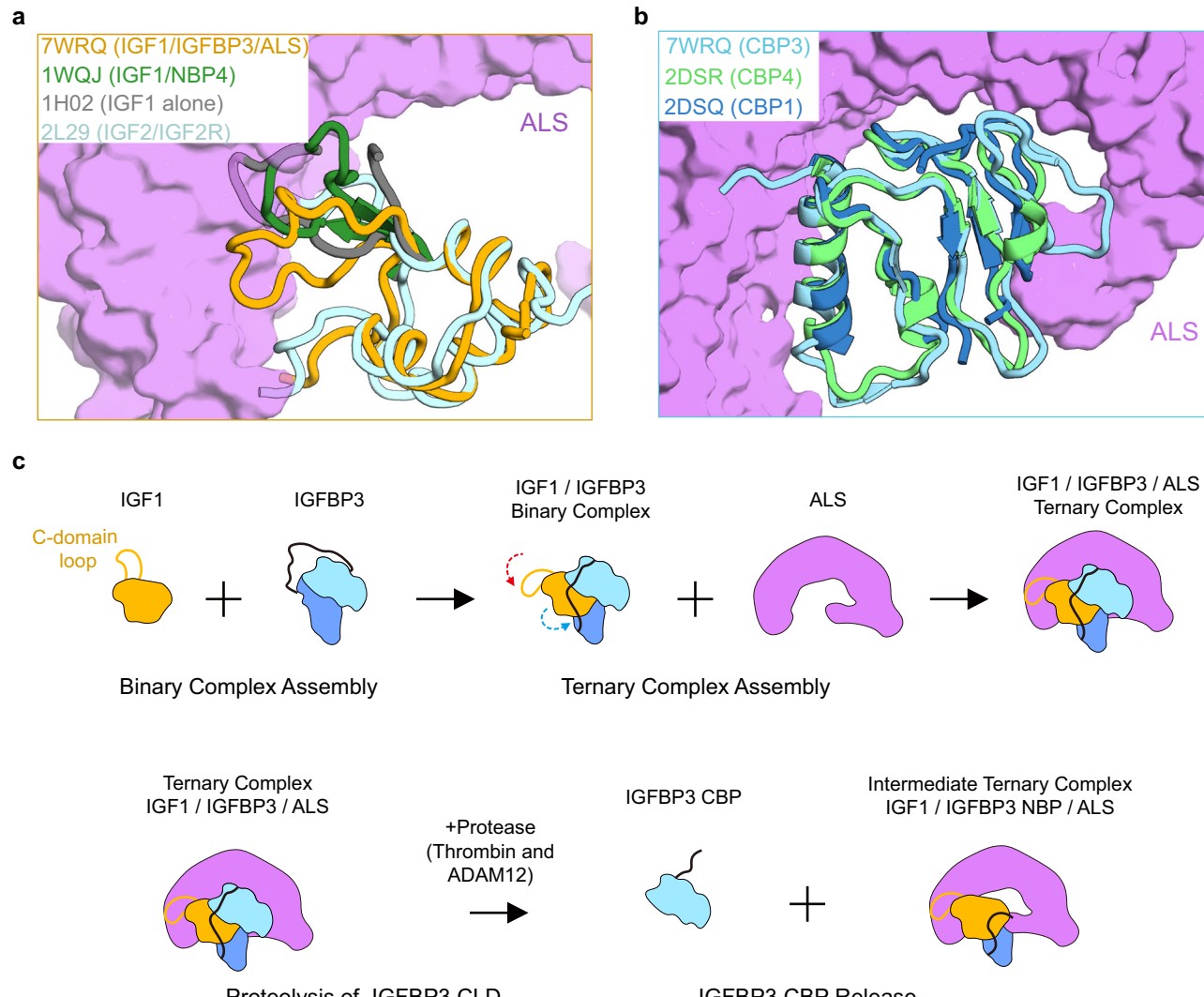

**Fig. 7 | Key determinants for specific assembly of the ternary complex and a proposed model for their assembly and disassembly. a** Comparison of IGF1 of the IGF1/IGFBP3/ALS ternary complex (yellow) with IGFs in a previously reported structure of free IGF1 [gray, PDB: 1H02[43]], IGF1/NBP4 complex [green, PDB: 1WQJ[21]], and IGF2/IGF2R complex [cyan, PDB: 2L29[66]]. Structures are shown as tube diagrams. For clarity, only the C domain of IGF1 in the binary complex (1WQJ) and IGF1 alone (1H02) are presented after superposition of these IGF1 structures. Residues missing in previous structures of IGF1 due to their flexibility are not shown. The orientation of the view of IGF1 and ALS is identical to that in Fig. 3d. **b** Comparison of CBP3 of the IGF1/IGFBP3/ALS ternary complex (cyan) with CBPs in the previously reported structure of IGF1/NBP4/CBP4 (lime, PDB: 2DSR) and IGF1/NBP4/CBP1 (blue, PDB: 2DSQ)[12]. Structures are shown as tube diagrams. The orientation of the view of CBP and ALS is identical to that in Fig. 3f. **a, b** ALS is presented as a violet surface. **c** Cartoon representation of the working model for assembly of the IGF1/IGFBP3/ALS ternary complex and formation of the intermediate ternary complex upon CLD proteolysis. IGF1 and ALS are yellow and purple, respectively. NBP and CBP of IGFBP3 are blue and cyan, respectively. The C-domain loop of IGF1 and CLD of IGFBP3 are indicated as lines.

or ADAM12 can cause subsequent dissociation of CBP from the ternary complex (Fig. 7c, bottom). It is still unclear how the CLD3 of uncomplexed IGFBP3 sterically blocks the ALS-binding interface and how the cleaved CLD3 (or CLD5) triggers dissociation of CBP3 (or CBP5) from the ternary complex. Further experiments, such as high-speed atomic force microscopy analysis and time-resolved cryo-EM of the intermediate ternary complex, are needed to ascertain the precise dynamics of IGFBP CLD in the binary and ternary complexes.

In conclusion, our comprehensive investigation of IGF1/IGFBP3/ALS ternary complexes provides insight into their sequential and specific assembly as well as CLD proteolysis-mediated intermediate ternary complex formation for IGF1R activation. Given the crucial role of the IGF axis in growth and proliferation and the genetic defects in IGF and ALS associated with various human diseases, our study also can pave the way for identifying eventual therapeutic targets of IGF axis dysfunction.

## Methods

### Cell lines and cell culture
HEK293A cell (#R70507, Thermo Fisher Scientific) were maintained in Dulbecco's Modified Eagle Medium (DMEM; #LM 001-05, Welgene) supplemented with 10% fetal bovine serum (FBS; #12484020, Thermo Fisher Scientific) and 1% Antibiotic-Antimycotic (#15240062, Thermo Fisher Scientific) and cultured at 37 °C in a humidified 5% $CO_2$ incubator. Expi 293-F cells (A14527, Thermo Fisher Scientific) were maintained in Expi293 Expression Medium (A1435102, Thermo Fisher Scientific) and cultured in a humidified shaking incubator at 37 °C and 8% $CO_2$.

### Constructs for recombinant protein expression and mutagenesis
Human IGF1 (residues M1-A118) followed by a stop codon, thrombin cleavage sequence, and Protein A-tag or Twin-Strep-tag were cloned

into the HindIII and XhoI sites of a pcDNA3.1 vector (#V79020, Invitrogen). For the GFP-fused IGF1 construct, human IGF1 followed by a GGGGs linker, EGFP (M1-239K), and Twin-Strep-tag was cloned into the HindIII and XhoI sites of the pcDNA3.1 vector. Human IGFBP3 (residues G28-K291) followed by a stop codon or Twin-Strep-tag, and human ALS (residues A28-C605) followed by a stop codon or 6x His-tag were cloned into the BamHI and NotI sites of the pcDNA 3.1 vector containing the signal sequence for protein secretion. To generate recombinant CLD3-CBP3, the CLD of IGFBP3 (residues S118-E209), CBP of IGFBP3 (residues 210Y-291K), and Twin-Strep-tag were cloned into the BamHI and NotI sites of the pcDNA 3.1 vector containing the signal sequence for protein secretion and Twin-Strep-tag. To generate recombinant hybrid-BPs, we performed overlap PCR using PCR fragments of the NBP and CBP of IGFBP3 and CLD of each human IGFBP [IGFBP1 (residues S108-W172), IGFBP2 (residues R135-A223), IGFBP4 (residues A104-P170), IGFBP5 (residues K104-E188), or IGFBP6 (residues R108-E159)] and subsequently cloned them into the BamHI and NotI sites of the pcDNA 3.1 vector containing the Twin-Strep-tag. Constructs and primer information are listed in Supplementary Tables 5 and 6. Mutants of IGF1, IGFBP3, and ALS at interacting interface were constructed from IGF1-Strep, IGFBP3, and ALS-His, respectively, using QuickChange PCR methods.

## Expression and purification of the IGF1/IGFBP3/ALS ternary complex for cryo-EM

A total of 1 µg of plasmid DNA (IGF1-proteinA: IGFBP3: ALS ratio of 1:2:4) was transfected into $2.5 \times 10^6$ Expi293-F cells (#A14527, Thermo Fisher Scientific) using Expifectamine (#A14524, Thermo Fisher Scientific), and cells were cultured in Expi293 expression medium (#A14351, Thermo Fisher Scientific) at 37 °C and 8% $CO_2$ with shaking (orbital shaker, 120 rpm) for 4 days. After centrifugation to remove the cells, the culture supernatant was loaded onto IgG Sepharose 6 Fast Flow (#17-0969-01, Cytiva). After washing with 10 column volumes of wash buffer (20 mM Tris-HCl pH 8.0, 200 mM NaCl), the Protein A–fused ternary complex (IGF1-protein A/IGFBP3/ALS)-bound resin was incubated with thrombin (1% [v/v] in wash buffer) at room temperature for 2 h to remove the C-terminal Protein A tag. The eluted IGF1/IGFBP3/ALS complex was concentrated to 0.8 mg/ml using an Amicon Ultra centrifugal filter (#UFC8030, Millipore) and further purified by SEC using a Superdex 200 Increase 10/300 GL column (#GE28-9909-44, Cytiva) equilibrated with a wash buffer (20 mM Tris-HCl pH 8.0, 200 mM NaCl). The peak fractions were concentrated to ~0.6 mg/ml using an Amicon Ultra centrifugal filter for cryo-EM analyses.

## Cryo-EM sample preparation and data collection

Quantifoil R1.2/1.3 300 mesh gold holey carbon grids (Cat#-Q3100AR1.3, Electron Microscopy Sciences) was glow-discharged with a PELCO easiGlow Glow Discharge Cleaning system (Ted Pella) for 90 s at 15 mA, and then 2.5 µl of the purified IGF1/IGFBP3/ALS complex was applied to the grid in 100% humidity at 4 °C. After 2 s of blotting, the grid was plunged into liquid ethane using a Vitrobot MkIV (Thermo Fisher Scientific). Micrographs were acquired on a Titan Krios G4 TEM operated at 300 keV with a K3 direct electron detector (Gatan) at the Institute for Basic Science (IBS), using a lit width of 20 eV on a GIF-quantum energy filter. EPU software was used for automated data collection at a calibrated magnification of ×105,000 under the single-electron counting mode and correlated-double sampling (CDS) mode[49], yielding a pixel size of 0.883 Å/pixel. A total of 14,754 movies were collected with a defocus range from −0.9 to −2.7 µm. Each micrograph was dose-fractionated to 42 frames under a dose rate of 14.5 e⁻/pixel/s with total exposure time of 3.4 s, resulting in a total dose of about 63 e⁻/Å². Detailed image acquisition parameters are summarized in Supplementary Table 1.

## Image processing, model building, and refinement

The detailed image processing workflow and statistics are summarized in Supplementary Fig. 1b and Supplementary Table 1. Raw movies were motion-corrected using MotionCorr2[50], and the CTF parameters were estimated by CTFFIND4[51]. All other image processing was performed using cryoSPARC v.3[52]. Initially, a total of 1,030,000 particles were picked with a blob picker of cryoSPARC from a few micrographs. 2D class averages representing projections in different orientations selected from the initial 2D classification were used as templates for automatic particle picking from the full datasets. The resulting 16,780,000 extracted particles were binned two times, and subsequent 2D classification, Ab initio, and Heterogeneous refinement for clean-up were performed in cryoSPARC. A total of 638,058 particles from the 3D classes showing good secondary structural features were selected and re-extracted into the original pixel size (300 × 300 pixel boxes). Non-uniform refinement was performed on the resulting volume using per-particle defocus and per-group CTF optimizations applied[53,54]. The final refinement yielded a map at an overall 3.2 Å resolution (with tight mask). The mask-corrected Fourier shell correlation (FSC) curves were calculated in cryoSPARC, and reported resolutions were based on the gold-standard Fourier shell correlation (FSC) = 0.143 criterion[55]. Local resolutions of density maps were estimated by Blocres[56].

Model building for IGFBP3 and IGF1 was initiated by docking the crystal structure of IGF1/IGFBP4 complex (PDB 2DSR [https://www.wwpdb.org/pdb?id=pdb_00002dsr]) into the cryo-EM map using the Phenix package (v1.19.2)[57]. The C-alpha chain and side chains of ALS were manually built in the cryo-EM density map using Coot (v0.8.9.2)[58]. The model was manually adjusted in Coot and refined against the map by using the real space refinement in the Phenix package. Model building of ambiguously resolved parts was aided by a protein model generated from AlphaFold2[59,60] and a post-processed map generated from DeepEMhancer[61]. The refinement statistics from Phenix validation are summarized in Supplementary Table 1. All figures for cryo-EM maps and structural models were generated using UCSF Chimera (v.1.1.4), ChimeraX (v.1.3), and PyMOL(v2.3.1)[62–64].

## Co-expression and pull-down assays

A total of 1 µg of plasmid DNA (ALS-His:IGFBP3-Strep:IGF1 = 1:1:2, ALS-His:IGFBP3-Strep = 1:1, ALS-His:IGF1-Strep = 1:1, ALS-His:CLD3-CBP3-Strep = 1:1, CLD3-CBP3-Strep:IGF1-His = 1:1, ALS-His:CLD3-CBP3-Strep:IGF1 = 1:1:2, ALS-His:hybrid-BPs-Strep:IGF1 = 1:1:2, or ALS-His:hybrid-BPs-Strep = 1:1) were transfected into $2.5 \times 10^6$ Expi293-F cells using Expifectamine, and cells were cultured in Expi293 expression medium at 37 °C and 8% $CO_2$ with shaking (120 rpm) for 4 days. Similarly, a total of 1 µg of plasmid DNA [ALS-His (mutant or WT):IGFBP3 (mutant or WT):IGF1-strep (mutant or WT) = 1:1:2] were transfected for complex formation analysis of mutants. Cells then were pelleted by centrifugation and the supernatants loaded onto Ni-NTA Agarose (#30210, Qiagen). The protein-bound resins were washed with 10 column volumes of wash buffer (20 mM Tris-HCl pH 8.0, 200 mM NaCl, 20 mM imidazole), and proteins were eluted with elution buffer (20 mM Tris-HCl pH 8.0, 200 mM NaCl, 300 mM imidazole). For Strep-Tactin purification, 10× buffer W (#2-1003-100, iba) and BioLock (#2-0205-050, iba) were added to supernatants and incubated for 20 min with agitation at 4 °C to prevent unwanted biotinylated proteins from binding to Strep-Tactin. After centrifugation (3134 x g, 20 min), the supernatants were loaded to Strep-Tactin Sepharose (#2-1201-025, iba) and washed with 10 column volumes of wash buffer (20 mM Tris-HCl pH 8.0, 200 mM NaCl). Then, proteins were eluted with elution buffer (20 mM Tris-HCl pH 8.0, 200 mM NaCl, 5 mM desthiobiotin).

## In vitro reconstitution of the binary or ternary complex and FSEC after proteolysis

To test the complex formation, the purified IGF1-GFP-Strep/IGFBP3 or IGF1-GFP-Strep alone were incubated with purified ALS (molar ratio of

1:1) for 16 h at 4 °C with shaking. Also, the purified IGFBP3-Strep or CLD3-CBP-Strep were incubated with purified IGF1-GFP-Strep (molar ratio of 1:1) for 16 h at 4 °C with shaking. Then, the mixtures were subjected to SEC using a Superdex 200 increase 10/300 GL column (#GE28-9909-44, Cytiva) equilibrated with 20 mM Tris pH 8.0, 200 mM NaCl. The flow path from the SEC column was directly connected to a Fluorescence Detector (FP-4025, JASCO) to detect the GFP fluorescence signal (488 nm/507 nm). To trace IGF1-GFP after proteolysis, the purified IGF1-GFP/IGFBP3 binary complex (Fig. 6a), IGF1-GFP/IGFBP3/ALS ternary complex (Fig. 6a and Supplementary Fig. 7a, c), or IGF1-GFP/hybrid-BPs/ALS ternary complexes (Fig. 6g and Supplementary Fig. 7i) were incubated with bovine thrombin (#91-035-020, Bio-Pharm, 0.2 unit of thrombin for 1 μg of IGF1-GFP/IGFBP3 binary complex, IGF1-GFP/IGFBP3/ALS or IGF1-GFP/hybrid-BPs/ALS ternary complexes in 20 mM Tris pH 8.0, 200 mM NaCl for 10 h at 37 °C with shaking), human ADAM12 (#4416-AD-020, R&D Systems, 0.2 μg of ADAM12 for 1 μg of IGF1-GFP/IGFBP3/ALS or IGF1-GFP/hybrid-BPs/ALS ternary complexes in 20 mM Tris pH 8.0, 200 mM NaCl, 10 mM $CaCl_2$ for 16 h at 37 °C with shaking) or human PAPP-A2 (#1668-ZN-020, R&D Systems, 0.2 μg of PAPP-A2 for 1 μg of IGF1-GFP/IGFBP3/ALS or IGF1-GFP/hybrid-BPs/ALS ternary complexes in 20 mM Tris pH 8.0, 200 mM NaCl for 16 h at 37 °C with shaking), followed by FSEC. To trace CBP3-GFP after proteolysis, the purified IGF1/IGFBP3-GFP/ALS ternary complex (Fig. 6c and Supplementary Fig. 7d) was incubated with bovine thrombin (#91-035-020, BioPharm, 0.2 unit of thrombin for 1 μg of IGF1/IGFBP3-GFP/ALS in 20 mM Tris pH 8.0, 200 mM NaCl for 10 h at 37 °C with shaking), human ADAM12 (#4416-AD-020, R&D Systems, 0.2 μg of ADAM12 for 1 μg of IGF1/IGFBP3-GFP/ALS in 20 mM Tris pH 8.0, 200 mM NaCl, 10 mM $CaCl_2$ for 16 h at 37 °C with shaking) or human PAPP-A2 (#1668-ZN-020, R&D Systems, 0.2 μg of PAPP-A2 for 1 μg of IGF1/IGFBP3-GFP/ALS in 20 mM Tris pH 8.0, 200 mM NaCl for 16 h at 37 °C with shaking), followed by FSEC. Also, IGF1/hybrid-BPs-GFP/ALS ternary complexes (Fig. 6h and Supplementary Fig. 7j) were incubated with bovine thrombin (#91-035-020, BioPharm, 0.2 unit of thrombin for 1 μg of IGF1/hybrid-BPs-GFP/ALS in 20 mM Tris pH 8.0, 200 mM NaCl for 10 h at 37 °C with shaking), human ADAM12 (#4416-AD-020, R&D Systems, 0.2 μg of ADAM12 for 1 μg of IGF1/hybrid-BPs-GFP/ALS in 20 mM Tris pH 8.0, 200 mM NaCl, 10 mM $CaCl_2$ for 16 h at 37 °C with shaking) or human PAPP-A2 (#1668-ZN-020, R&D Systems, 0.2 μg of PAPP-A2 for 1 μg of IGF1/hybrid-BPs-GFP/ALS in 20 mM Tris pH 8.0, 200 mM NaCl for 16 h at 37 °C with shaking), followed by FSEC. IGF-GFP/IGFBP3/ALS, IGF1-GFP/IGFBP3, and IGF1-GFP were detected by fluorescence signal, but ALS was detected with a UV/vis detector.

### Immunoblotting for analysis of ternary complex components after thrombin proteolysis

After incubation of the purified IGF1/myc-IGFBP3-Strep/ALS ternary complex with bovine thrombin (#91-035-020, BioPharm, 0.2 unit of thrombin for 1 μg of IGF1-GFP/IGFBP3/ALS) in 20 mM Tris pH 8.0, 200 mM NaCl for 10 h at 37 °C with shaking (Fig. 6b), the samples were subjected to SEC using a Superdex 200 increase 10/300 GL column (#GE28-9909-44, Cytiva) equilibrated with 20 mM Tris pH 8.0, 200 mM NaCl. The peak fractions were treated with sodium dodecyl sulfate (SDS) sample buffer and subjected to denaturing polyacrylamide gel electrophoresis (SDS-PAGE). Proteins were transferred to a polyvinylidene fluoride (PVDF) membrane (#88518, Thermo Fisher Scientific). The membrane was blocked by incubation with 5% (w/v) skim milk in Tris-buffered saline containing 0.1% Tween-20 (TBS-T) at room temperature (RT) for 1 h, and then incubated with primary antibodies [Myc-tag (1:1000 dilution, #2278 s, Cell Signaling); Strep-tag II (1:1000 dilution, #ab76949, Abcam); IGFBP3 (1:1000 dilution, #25864s, Cell Signaling); IGFBP3 (1:1000 dilution, #sc-374365, Santa Cruz); ALS (1:1000 dilution, #sc-377131, Santa Cruz); IGF1 (1:1000 dilution, #MA1-088, Thermo Fisher)] at 4 °C for 12 h. After being washed with TBS-T buffer three times, the membrane was incubated at

RT for 30 min with horseradish peroxidase (HRP)-conjugated anti-mouse antibodies (#62-6520, Thermo Fisher Scientific) for anti-IGFBP3 (#sc-374365), anti-ALS, and anti-IGF1 primary antibodies, or HRP-conjugated anti-rabbit antibodies (#31460, Thermo Fisher Scientific) for anti-Myc-tag, anti-Strep-tag II, and anti-IGFBP3 (#25864s) primary antibodies.

### IGF1R activation assay

HEK293A cells were cultured in DMEM (001-05, Welgene) with 10% FBS at 37 °C and 5% $CO_2$ in 6-well plates. Cells were starved by incubation in DMEM for 12 h, after which they were treated for 3 min with 5 nM of IGF1, ternary complex (IGF1/IGFBP3/ALS or IGF1/hybrid-BPs/ALS), or the SEC fraction of the ternary complex (IGF1/IGFBP3/ALS or IGF1/hybrid-BPs/ALS) after proteolysis with bovine thrombin (#91-035-020, BioPharm, 0.2 unit of thrombin for 1 μg of ternary complex in 20 mM Tris pH 8.0, 200 mM NaCl for 10 h at 37 °C with shaking), human ADAM12 (#4416-AD-020, R&D Systems, 0.2 μg of ADAM12 for 1 μg of ternary complex in 20 mM Tris pH 8.0, 200 mM NaCl, 10 mM $CaCl_2$ for 16 h at 37 °C with shaking) or human PAPP-A2 (#1668-ZN-020, R&D Systems, 0.2 μg of PAPP-A2 for 1 μg of ternary complex in 20 mM Tris pH 8.0, 200 mM NaCl for 16 h at 37 °C with shaking) (Fig. 6d, e, f and Supplementary Fig. 7b, e, g, h). Thereafter, cells were washed twice with cold PBS and lysed in 100 μl of lysis buffer (50 mM Tris-HCl pH 8.0, 150 mM NaCl, 1% Triton X-100, protease inhibitor, phosphatase inhibitor). After incubation at 4 °C for 60 min, cell lysates were cleared by centrifugation at $15,000 \times g$ for 20 min, after which the protein concentration in the supernatant was quantified by the Bradford assay. Then, the cell lysates were treated with SDS sample buffer and subjected to SDS-PAGE. Proteins were transferred to a PVDF membrane (#88518, Thermo Fisher Scientific). IGF1R phosphorylation was assessed by first blocking the blot by incubation with 5% (w/v) skim milk in TBS-T at RT for 1 h, and then incubating it with an anti-phospho-IGF1R primary antibody (Y1135/1136) (1:1000 dilution, #3024, Cell Signaling) at 4 °C for 12 h, followed by incubation with HRP-conjugated goat anti-rabbit IgG antibody (1:5000 dilution, #31460, Thermo Fisher Scientific). Immunoreactive phospho-IGF1R was visualized using EzWestLumiOne (#2332632, ATTO). After staining, the membrane was incubated in stripping buffer (Thermo Fisher Scientific) for 15 min and re-probed with a rabbit anti-IGF1 receptor antibody (1:1000 dilution, #3027, Cell Signaling), followed by incubation with HRP-conjugated goat anti-rabbit IgG antibody (1:5000 dilution, #31460, Thermo Fisher Scientific) to determine the amount of total IGF1R. Another blocked membrane was incubated with an anti-beta-actin (1:1000 dilution, #sc-4778, Santa Cruz) followed by incubation with HRP-conjugated goat anti-mouse IgG antibody (1:5000 dilution, #62-6520, Thermo Fisher Scientific). Immunoreactive IGF1R and beta-actin were visualized using EzWestLumiOne (#2332632, ATTO).

### Pull-down and Immunoblotting for analysis of intermediate ternary complex which binds to IGF1R

After incubation of the purified IGF1R ectodomain-leucine zipper-strep (IGR1R-zip) with IGF1, and intermediate ternary complex (IGF1/IGFBP3/ALS after thrombin proteolysis, 0.2 unit of thrombin for 1 μg of IGF1-GFP/IGFBP3/ALS in 20 mM Tris pH 8.0, 200 mM NaCl for 10 h at 37 °C with shaking), the mixture was loaded onto Strep-Tactin Sepharose (#2-1201-025, iba) and washed with 10 column volumes of wash buffer (20 mM Tris-HCl pH 8.0, 200 mM NaCl). Strep-Tactin resin and flow through sample were treated with sodium dodecyl sulfate (SDS) sample buffer and subjected to denaturing polyacrylamide gel electrophoresis (SDS-PAGE). Proteins were transferred to a polyvinylidene fluoride (PVDF) membrane (#88518, Thermo Fisher Scientific), which in turn was blocked by incubation with 5% (w/v) skim milk in Tris-buffered saline containing 0.1% Tween-20 (TBS-T) at room temperature (RT) for 1 h, and then incubated with primary antibodies [His-tag (1:1000 dilution,

# A00186, GeneScript); Strep-tag II (1:1000 dilution, #ab76949, Abcam); IGFBP3 (1:1000 dilution, #25864s, Cell Signaling); ALS (1:1000 dilution, #sc-377131, Santa Cruz)] at 4 °C for 12 h. After being washed with TBS-T buffer three times, the membrane was incubated at RT for 30 min with horseradish peroxidase (HRP)-conjugated anti-mouse antibodies (#62-6520, Thermo Fisher Scientific) for anti-ALS, and anti-His-tag primary antibodies, or with HRP-conjugated anti-rabbit antibodies (#31460, Thermo Fisher Scientific) for anti-Strep-tag II, and anti-IGFBP3 (#25864s) primary antibodies.

## Quantification and statistical analysis
Data in figures and tables are reported as mean ± standard deviation (SD) with the number of biological and technical replicates indicated in the figure and table legends, where "*n*" represents the number of biological replicates. Immunoblotting was assessed using ImageJ and GraphPad Prism v9.0.

## Reporting summary
Further information on research design is available in the Nature Research Reporting Summary linked to this article.

## Data availability
The atomic coordinates for the IGF1/IGFBP3/ALS ternary complex have been deposited to the RSCB Protein Data Bank under the accession number 7WRQ. The cryo-EM map of the ternary complex has been deposited on Microscopy Data Bank under the accession number EMD-32735. The source data for all figures and supplementary figures are available as Source Data file. Reporting Summary for this article is provided as Supplementary Information file. All the other data and materials used for the analysis are available from the corresponding author upon reasonable request. Source data are provided with this paper.

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

## Acknowledgements

We are grateful to the staff of the Research Solution Center at IBS for help with cryo-EM data collection. Computational work for this research was performed on the data analysis hub, Olaf, in the IBS Research Solution Center. We also thank Global Science experimental Data hub Center (GSDC) at Korea Institute of Science and Technology Information (KISTI) for computing resources and technical support, and Dr. Jin Young Kang, Jinwoo Kim at KAIST, Department of chemistry for image processing counsel. This work was supported by grants from the Institute for Basic Science (IBS-R030-C1 to H.M.K.).

## Author contributions

H.K., Y.F., and H.M.K. designed the experiments and analyzed the data; H.J.H. and D.S.L. purified proteins; H.K., Y.F., and S.-G.L. determined the cryo-EM structure; and H.K., Y.F., and H.M.K. wrote the manuscript.

## Competing interests

The authors declare no competing interests.
