## [Peer Review File · Nature Communications]

Structural basis for assembly and disassembly of the IGF/IGFBP/ALS ternary complexREVIEWER COMMENTS

Reviewer #1 (Remarks to the Author):

IGFBP3 or IGFBP5 together with ALS protect the degradation of IGF1 in circulation through forming a stable 1:1:1 ternary complex. Kim et. al here presents the cryo-EM structure of IGF1/IGFBP3/ALS ternary complex, which is a long-awaited structure in IGF field. This structure shows how IGF1, IGFBP3 and ALS are assembled in good detail. Combined with a large body of biochemical experiments, the authors also illustrate the inhibitory role of the CLD domain of IGFBP3 in the complex formation, reveals the process of sequential assembly of the ternary complex, and provides structural explanation for why IGF1/IGFBP3 and IGF1/IGFBP5 but not other IGF1/IGFBPs can interact with ALS. Overall, this is a nice piece of work, and the structural information present here is certainly valuable. However, there are a few major issues need to be addressed before this work could be published. This includes: (1) the resolution for the cryo-EM map is overestimated to some extent. (2) The structural model is not validated by mutagenesis and biochemical experiments. (3) The proposed model that IGF1/IGFBP3-NBP/ALS could activate IGF1R without dissociation is very speculative due to the lack of strong structural and biochemical evidences. Substantial amount of work is required for the revision. My specific points are:

(1) The overall resolution for the cryo-EM map is clearly overestimated. The resolution should be calculated based on the FSC between map and model at 0.5 (not 0.143). Based on the model/map FSC curve shown in Extended Figure 1d, the resolution is close to 4 Å, rather than 3 Å that is reported in the paper based map/map FSC. The inconsistent resolution estimation result between using different criteria is due to the over-fitting. The authors should optimize the 3D refinement to prevent the overfitting. The author could try to apply different programs (cryoSPARC versus RELION) to refine the structure.

(2) Related to my last point. Since the resolution for the cryo-EM map is not very high, the reliability of the model building becomes questionable, especially for ALS whose structure is not reported before. To convince the readers that the model is accurate, the authors need to explicitly show the density of each domain of IGF1/IGFBP3/ALS with the model fitted. It is also important to show the densities for glycosylation and disulfide bonds of ALS. This is a good way to validate the structure. Some key regions of ALS were resolved at even lower resolution, e.g., the hook loop of ALS. The authors need to elaborate in the methods section about how the model building is performed with the poor cryo-EM densities.

(3) The author didn't perform any mutagenesis and binding experiment to validate the structural model of 1:1:1 IGF1/IGFBP3/ALS ternary complex. It is, no doubt, that such validation is required when the complex structure is determined for the first time. Therefore, the authors need to introduce some mutations to each interface between IGF1 and ALS and between IGFBP3 and ALS, and test the effect

these mutation on the formation of ternary complex by using the pull-down experiments. The interface between IGF1 and IGFBP3 doesn't need to be validated as similar structure has been determined before.

(4) The authors proposed that, after the proteolysis, CBP of IGFBP3 would fall apart from the ternary complex and the remaining IGF1/IGFBP3-NBP/ALS could activate IGF1R without further dissociation. Such claim is too speculative, because it is only supported by a cell-based auto-phosphorylation experiment. As IGF1 is localized in the inner surface of half-moon shaped ALS, it is hard to imagine how IGF1/IGFBP3-NBP/ALS could bind the L1/ α CT of IGF1R (IGF1 binding site in IGF1R) without causing any clash between ALS and IGF1R. In addition, previous structure has shown that IGF1 is sandwiched between two protomers in the active state of IGF1R. It is impossible that IGF1/IGFBP3-NBP/ALS could stabilize the active state of IGF1R in a similar fashion as IGF1 does.

Here is my alternative explanation for the functional experiment result. After CBP of IGFBP3 falls apart from ternary complex by proteolysis, the remaining IGF1/IGFBP3-NBP/ALS complex will become relatively instable. It is therefore very likely that IGF1 binds to IGFBP3-NBP/ALS complex in a much lower affinity than to

IGF1R (K_d : ~ 1 nM). As a result, the presence of IGF1R would shift the association-dissociation equilibrium of IGF1/IGFBP3-NBP/ALS complex, thereby promoting the quick dissociation of IGF1/IGFBP3-NBP/ALS complex. The released IGF1 from ternary complex is able to activate IGF1R on its own. If the authors want to make a strong claim that IGF1/IGFBP3-NBP/ALS as a whole can bind and activate IGF1R, they need to provide more structural and biochemical evidences. For instance, they could solve the cryo-EM structure of IGF1/IGFBP3-NBP/ALS in complex with IGF1R to confirm that IGF1/IGFBP3-NBP/ALS could bind IGF1R altogether. Such structure would be informative even at low resolution. They could also perform rigorous in vitro binding experiment to show that IGF1/IGFBP3-NBP/ALS could bind IGF1R without the requirement of dissociation.

(5) Related to my last point. In Fig.6a, small amount of IGF1 is certainly released from ternary complex after proteolysis. How long does the authors incubate IGF1-GFP/IGFBP3/ALS with thrombin in this experiment? The authors need to perform a time course experiment (like that in Fig.6f) to make sure that the amount of free IGF1 doesn't increase with longer proteolysis time.

Reviewer #2 (Remarks to the Author):

This manuscript reports a 3 Å resolution cryo EM structure of the IGF1:IGFBP-3:acid-labile subunit complex. Whilst the structure of ALS has been predicted for some time through homology modelling,

this is the first structure of ALS and the first structure of the ternary complex. The ternary complex structure reveals the surfaces of interaction between the three components and confirms previous predictions. Molecular detail of specific sites of interaction between ALS and IGFBP-3 and ALS and IGF1 are identified. The structural information is used to explain effects of clinically relevant mutations. Through a series of pulldown and size exclusion chromatography experiments with the ternary complex and differently tagged components some interesting observations regarding the assembly and proteolytic release of IGF1 are made. Firstly, the data confirm previous experiments that showed the ternary IGF1:IGFBP-3:ALS complex requires prior formation of the binary IGF1:IGFBP-3 complex. Interestingly if the linker-domain is replaced by a linker-domain from either IGFBP-1, -2, -4 or -6 a complex between the hybrid receptors and ALS can be formed, implying that the linker-domains of IGFBP-3 and -5 have unique features that prevent binding unless pre bound to IGF1. Whether the

Somewhat controversially the authors also suggest that the proteolysis of the IGFBP3 linker domain in the ternary complex induces a linker-domain conformational change and subsequent dissociation of BP3 C-domain from the ternary complex. The suggestion is that a IGF1 remains in the high molecular weight complex and a hydrophobic patch on IGF1 is exposed that is critical for IGF1R binding, rather than the current understanding that proteolysis results in release of free IGF1 that then binds and activates the IGF-1R. To definitively show this it would be good to isolate the digested complex from CBP-3 and test whether this is able to activate the IGF-1R. Is it possible to predict whether the entire complex with the exposed IGF1 hydrophobic patch could be accommodated in the IGF-1R binding site as suggested?

This manuscript represents a significant body of data that provides insight into the mechanism of action of the ternary IGF1:IGFBP-3:ALS complex.

Specific comments/corrections

Page 2 first sentence "The structure of the IGF1/IGF1R complex also explains why, as compared with IGF1, IGF2 and insulin with short C-domain loops exhibit weaker binding to IGF1R" This sentence is inaccurate or ambiguous. The mature insulin does not have a C-domain. Also it would be better to say that the structure of the IGF1/IGF-1R complex reveals why the IGFs are unable to bind the IGF-1R if they exist in an IGF:IGFBP binary complex. The binding pocket is too small to accommodate the binary complex and the IGF residues involved in the IGF-1R primary high affinity binding site are engaged in the interaction with the IGFBP, thereby preventing binding to the receptor.

Page 4 last paragraph "the IGF1/IGFBP3 binary engages" should be "the IGF1/IGFBP3 binary complex engages"

Page 5 end of discussion. It is not evident what the “intermediate ternary complex that is crucial for IGF1R activation” is. It has not been defined until the discussion.

Page 7 second sentence. “nine highly conserved disulfide bonds (3 in IGF1, 6 in NBP3, and 3 in CBP3) crucial for rigidity of the core modules.” There are 12 disulfide bonds described here in brackets. Which 9 are you referring to?

It would be good to define which residues of the N domain form the “thumb” structure as this region is discussed at length in the results.

IGF1 α B helix is not a common terminology – perhaps describe as the IGF1 B domain α helix. Similarly it is not clear what the IGF1 Nterminal loop (L5) is – please define. The term IGF1 α B wedge helix is not common and needs to be defined. The terminology “CBP3 β 4 and IGF1 β B” is not defined.

End of page 8. Perhaps better worded as “the thumb region of NBPs is of different length (NBP3, 12; NBP5-6aa, NBP1-4aa, NBP2-6aa, NBP4-5aa, NBP6-1aa amino acids long)”

It is not clear if the binary complex being described is just the complex within the ternary complex or whether binary complex particles were detected when collecting data for the ternary complex. There is no specific methods section describing generation of the binary complex for cryo EM and the collection of binary complex cryo EM data. No particle data is provided.

Page 11: “the mutation on Leu and Asp residues” should read “the mutation on Leu and Asn residues”

“Although IGF1 R36Q and V44M are located at the IGF1 interaction interface with ALS and CBP3, respectively, these mutations do not seem to affect IGF1 binding to ALS or IGFBP3 (Fig. 4e).” – authors should refer to “Netchine et al (2009) DOI 10.1210/jc.2009-0452” and “Walenkamp MJ et al (2005) DOI: 10.1210/jc.2004-1254 which demonstrate that R36Q and V44M IGF-I can still bind IGFBP-3, respectively, and V44M is also able to assemble into the ternary complex (Walenkamp et al).

Surprisingly, we found that even in the absence of IGF1, ALS could bind to hybrid-BP1, -2, -4, and -6, but not to IGFBP3 or hybrid-BP5

What is the affinity of the hybridBPs for ALS? Rather than a structural change in the linker upon IGF1 being the mechanism by which BP-3 and BP-5 can then bind ALS, could this be directly due to differences in amino acid sequences? There are a lot of basic residues in BP-3 and BP-5 linker domains.

Fig 6 legend. It is not clear how long the thrombin cleavage was conducted for in fig a, g and h. Please include this in the legend.

Is there a possibility that in the time frame of the thrombin cleave in fig 6g that there could be some proteolysis leading to removal of the GFP from IGF1?

Reviewer #3 (Remarks to the Author):

Reviewer Comments to Author

The MS entitled “Structural basis for assembly and disassembly of the IGF/IGFBP/ALS ternary complex” by Hyojin Kim, Yaoyao Fu, Ho Jeong Hong, Seong-Gyu Lee, Dong Sun Lee, and Ho Min Kim from the School of Medical Science and Engineering, and the Advanced Institute of Science and Technology (KAIST), both from the Republic of Korea is a very interesting study aiming to understand the structural complexity of the IGF/IGFBP/ALS ternary complex.

Interestingly, this is the first report of the spatial structure of ALS by using Cryo-EM to overcome the difficulties to obtain a crystalized ALS protein suitable for X-ray diffraction studies. In addition, the spatial structure for the entire IGF/IGFBP/ALS ternary complex was also resolved by image processing with an acceptable overall resolution of 3.0 Å.

The authors also characterize the crucial determinants for the sequential assembly of IGF1 with IGFBP3, and then the binary complex with ALS. Finally, a series of elegant in vitro experiments show that in the absence of IGF1, the central linker domain (CLD) of IGFB3 blocks the interaction with ALS and that the proteolysis of CLD (at least when thrombin was the selected protease) is crucial for the availability of IGF1 for interaction with its receptor.

The MS is well written, and the findings are clearly presented, however, there are several points that should be clarified.

Abstract

. . . suggesting a structural basis for human diseases associated with IGF, IGFBP3, and ALS, such as complete ALS deficiency (ACLSLSD) and Laron syndrome.

The associated human diseases the authors want to refer to are the complete ALS deficiency (ACLSLSD; OMIM #615961) and IGF1 deficiency (OMIM #608747). The Laron syndrome is caused by inactivating mutations/deletions in the GH receptor and the associated IGF1 deficiency results from the impairment of GH action. Indicating the OMIM numbers would be useful for the readers to access to the related human diseases that result from genetic defects in the genes coding for the members of the IGF/IGFBP/ALS ternary complex, mainly IGF1 and IGFBP3, since so far, no human mutation has been reported in the IGFBP3 gene.

I would like to suggest changing the paragraph to read as follow:

. . . suggesting a structural basis for human diseases associated with IGF1 and IGFBP3 gene mutations, such as complete ALS deficiency (ACLSLSD, OMIM #615961) and IGF1 deficiency (OMIM #608747).

Main

Page 3

IGFs can form binary complexes (10%–15%) with IGF-binding proteins (IGFBPs: IGFBP1, IGFBP2, IGFBP4, or IGFBP6), or ternary complexes (80%–90%) with IGFBPs (IGFBP3 or IGFBP5) and acid-labile subunit (ALS)2-4. In the circulation, free IGFs have a half-life of less than 10 min, but a binary complex increases this half-life to 30–90 min, whereas a ternary complex maintains it up to 16–24 h5.

Change to read as follow:

In the circulation, IGFs can form binary complexes (10%–15%) with IGF-binding proteins (IGFBPs: IGFBP1, IGFBP2, IGFBP4, or IGFBP6), or ternary complexes (80%–90%) with IGFBPs (IGFBP3 or IGFBP5) and acid-labile subunit (ALS)2-4. In the circulation, Free IGFs have a half-life of less than 10 min, but a binary complex increases this half-life to 30–90 min, whereas a ternary complex maintains it up to 16–24 h5.

Page 4

Individuals with complete ALS deficiency (ACLSLSD) have mutations in the IGFBP3 gene and exhibit severely reduced serum IGF1 and IGFBPs (particularly IGFBP3), leading to low birth weight and length, reduced head circumference and height, pubertal delay, and insulin resistance^{15,16}.

I would like to suggest adding the reference of the original description of ALS deficiency and again the OMIM#.

ACLSL: OMIM #615961

Domene HM, N Engl J Med 350, 570-7 (2004).

Page 5

Taken together, our findings provide mechanistic insight into assembly of the IGF/IGFBP3/ALS ternary complex and its disassembly for IGF bioavailability and allowed us to interpret the structural effects of IGF1 and ALS mutations in ACLSD and Laron syndrome.

Change to:

Taken together, our findings provide mechanistic insight into assembly of the IGF/IGFBP3/ALS ternary complex and its disassembly for IGF bioavailability and allowed us to interpret the structural effects of IGF1 and ALS mutations in ACLSD and IGF1 deficiencies.

Page 8

It has been reported that the extended N-terminal thumb of NBP is critical not only for regulation of IGF1 binding to IGF1R but also for interactions between NBP and CBP.

The authors should add a reference to this assessment.

Page 11

Structural basis for disease-associated ALS and IGF1 mutations

To date, various mutations (18 missense, 7 frameshift, 2 in-frame insertion, 1 nonsense mutation, and 1 deletion of exon 2) in the human ALS gene have been reported in patients with ACLSD16 (Fig. 4a, b). Although the 18 missense mutations are located in the LRRNT (C60S and P73L), LRR (N84S, L127P, L134Q, T145K, L172F, L213F, L241P, L244F, N276S, L409F, D440N, A475V, D488N, and S490W), and LRRCT domains (C540R and P586L), none of the residues are involved in direct interactions with the IGF1/IGFBP3 binary complex.

As it was reported in reference #16 (Domene S & Domene HM, Mol Cell Endocrinol 518, 111006, 2020) 28 different mutations have been described in ACLSD (17 missense, 7 frameshift, 2 in-frame insertion, 1 nonsense, and 1 deletion of the entire exon 2). In some patients two variants have been found in cis (L409F and V475V). In vitro expression of these variants and functional ternary complex analysis demonstrates that while the A475V variant is benign, is able to be expressed, secreted and retains its ability for ternary complex formation, the L409F variant is not expressed in vitro and is the one responsible for the complete ACLSD of the patient (Martucci LC, et al. Mol Cell Endocrinol 429, 19-28, 2016).

The A475V should be eliminated from the text, to read as follow:

To date, various mutations (17 missense, 7 frameshift, 2 in-frame insertion, 1 nonsense mutation, and 1 deletion of exon 2) in the human ALS gene have been reported in patients with ACLSD16 (Fig. 4a, b).

Although the 18 missense mutations are located in the LRRNT (C60S and P73L), LRR (N84S, L127P, L134Q, T145K, L172F, L213F, L241P, L244F, N276S, L409F, D440N, A475V, D488N, and S490W), and LRRCT domains (C540R and P586L), none of residues are involved in direct interactions with the IGF1/IGFBP3 binary complex.

In addition, the reference from Martucci et al. should be added.

Figure 4

Figure 4 presents several inconsistencies. First, to the knowledge of this reviewer the variants C60F and the A434V have not been previously described. As was previously mentioned, the following variants: R277H, P287L, A475V, and R548W have been described in children with short stature but not associated to complete ACLSD (Domené HM et al. *Horm Res Paediatr* 80, 413-23, 2013). In addition, similarly to the A475V variant, P287L and R548W are normally expressed and secreted in vitro and retain their ability to form ternary complexes.

For all these reasons, variants C60F, R277H, P287L, A434V, A475V, and R548W, should be removed from both the schematic representations of Figure 4a and 4b (Martucci LC, et al. *Mol Cell Endocrinol* 429, 19-28, 2016).

Pages 11-12

In particular, the mutation on Leu and Asp residues (N84S, L127P, L134Q, L172F, L241P, L244F, N276S, and L409F) as part of the LRR consensus sequence (xLxxLxLxxNxLxxLxxxxFxx/Lx) are likely to introduce a main chain distortion or disrupt the LRR hydrophobic core, affecting the entire LRR architecture of the ALS protein. Similar to this effect, substitution of T145 and L213 residues to long charged Lys and bulky hydrophobic Phe, respectively, which point toward the hydrophobic core of the LRR solenoid interior, would introduce steric hindrance with surrounding residues and destroy the regular fold of the LRR module. The highly conserved C60 and C540 residues participate in disulfide bridge formation with C45 and C583, respectively, and the P73 and P586 residues contribute to maintaining structural rigidity. Therefore, mutations at these residues (C60S, C540R, P73L, and P586L) could destabilize LRRNT and LRRCT (Fig. 4c).

Although D440 and D448 are not essential to ALS/IGFBP3 interactions, D440N and possibly D488N could introduce a new N-glycosylation site that leads to impairment of secretion of ALS and its ability to form a ternary complex, as reported previously²². The S490W mutation is predicted to cause loss of the hydrogen bond network with neighboring S466, R467, D488, H491, S512, and R514 on the surface of the LRR17-LRR19 ascending flank (Fig. 4c). Two in-frame mutants, S195_R197dup and L437_L439dup, insert three amino acids at LRR6 and LRR16, respectively. The resulting altered length of the LRR modules might impair alignment of the hydrophobic residues on the consensus motif, perturbing the solenoid-like structure of ALS. Moreover, S195_R197dup and L437_L439dup lie adjacent to the ternary complex interface, probably disrupting ternary complex formation.

In the previous paragraph the authors analyze the effect of several IGFALS gene variants on the overall structure of the ALS protein. As they rightly mention, D440N and D488N could introduce a novel N-glycosylation site that may result in secretion impairment (this was already demonstrated by Firth SM et al. reference 22). The authors should also mention that David et al. (reference 19) already analyzed variants L134Q, L172F, L241P, L244F, N276S, C60S, C540R, P73L, S195_R197dup and L437_L439dup and according to their own predictive ALS spatial structure all of them resulted in major distortions in ALS structure. By using the same structural model, Martucci et al. also reported major distortion in L213F and L409F. *in vitro* expression of these ALS variants demonstrates that while L409F is not synthesized, L213F is partially expressed but is unable to be secreted remaining trapped inside the cells. These previous finding should be mentioned.

Page 12

IGF1 deficiency and several bio-inactive IGF1 mutations (R36Q, V44M, and R50W) also have been identified in patients with growth failure 23,24 (Fig. 4d). Although IGF1 R36Q and V44M are located at the IGF1 interaction interface with ALS and CBP3, respectively, these mutations do not seem to affect IGF1 binding to ALS or IGFBP3 (Fig. 4e). However, the decreased affinity of these IGF1 mutants for IGF1R may result in phenotypes such as severe intrauterine and postnatal growth retardation, microcephaly, and sensorineural deafness²³. Taken together, our findings provide a structural basis for ACLSD and the impaired growth seen with genetic defects in the human IGF1/IGFBP3/ALS ternary complex.

The authors omitted one reported IGF1 mutation which resulted in pre- and postnatal growth retardation, microcephaly, facial dysmorphism, bilateral sensorineural deafness, and mild global development delay (Keselman et al. *Europ J Endocrinol* 181, K43-K53, 2019). Tyr60 residue is highly conserved among species, and also conserved in IGF2 and insulin. Although the reported T60H would not affect the interaction with IGFBP3, T60 is a key residue involving in IGF1R binding (Bayne ML et al. *J Biol Chem* 265,15648-52, 1990).

These references should also be added, analyzed in the text, and reported in Figure 4d.

Page 14

Release of IGFBP3 CBP from ternary complex upon IGFBP3 CLD proteolysis is crucial for IGF1R activation
Proteolysis at the IGFBP CLD by proteases such as metalloproteinase, cathepsins, and serine proteases is crucial for activation of IGFR signaling in a target tissue or cell^{10,25}. To check whether IGFBP proteolysis causes IGF1 release from the ternary complex, we incubated the purified IGF1-GFP/IGFBP3 binary complex or IGF1-GFP/IGFBP3/ALS ternary complex with thrombin, which cleaves R97-A98 and R206-G207 of IGFBP3²⁵, and used FSEC to monitor IGF1-GFP release from the complex. As expected, IGF1-GFP was released from the binary complex after thrombin digestion and eluted at the peak position, corresponding to free IGF1-GFP (Fig. 6a). Strikingly, there was no free IGF1-GFP peak released from the ternary complex after thrombin digestion, in contrast to previous reports²⁶.

Several proteases are responsible for IGFBP3 (and IGFBP5) proteolysis. They have different proteolytic sites on the CLD3 resulting in different fragments and, potentially in different IGFBP3 fragments. Thrombin cleaves specifically the CLD of IGFBP3 at two sites: R97-A98 and R206-G207 (Booth BA et al. *Am J Physiol* 271, E465-70, 1996; Firth SM & Baxter RC *Endocr Rev* 23, 824-54, 2002). PAPP-A2 (a specific protease for IGFBP3 and IGFBP5) appears to constitute the most relevant physiologic protease present in the circulation. It has been shown that PAPP-A2 cleaves IGFBP5 at S143-K144 but no report indicating the specific digestion site for IGFBP3 has yet been reported. However, it is likely that the fragments resulting from PAPP-A2 proteolysis could have less ability to retain IGF1. This could explain the different results obtained by Lee & Rechler when they incubated ternary complex with adult rat serum and observed release of IGF1 after proteolysis of IGFBP3 that was then transferred to rat IGFBPs during the incubation (reference 26 in the MS). The physiological relevance of PAPP-A2 is further emphasized by the description of PAPP-A2 deficiency, as a result of inactivating mutations in the PAPP2 gene, characterized by short stature associated to high levels of IGF1, IGFBP3, IGFBP5 and ALS (Dauber A et al. *EMBO Mol Med* 8, 363-74, 2016) (OMIM# 619489).

Page 15

Of interest, the CBP3 band was not detected after thrombin digestion and subsequent SEC, whereas ALS, IGF1, and cleaved NBP3 remained in the ternary complex (Fig. 6b). These results indicate that CBP3, not IGF1, is released from the ternary complex after proteolysis at the CLD of IGFBP3.

There is no experimental data supporting that the lack of CBP3 in the complexes obtained after thrombin digestion can be extended to all other IGFBP3 proteases, particularly to PAPP-A2. It is possible that different proteases, having their private digestion sites, may produce different fragments of IGFBP3 retaining or not CBP3.

The authors should clarify this point.

Discussion

The authors should emphasize that the cryo-EM deduced ALS structure clearly sustains the previous model proposed by David et al. as a flat horseshoe-like, and completely different to those proposed by Janosi et al.

In addition, there are two important structural findings, reported in this manuscript, that were not observed in the David model: the long hook loop on LRRCT protruding into the central axis and the glycans attached to N368.

Page 19

It has long been known that IGFBP proteolysis of the linker domain leads to increased concentrations of bioavailable IGF and subsequent activation of the IGF1R37. Surprisingly, we found the intermediate ternary complex after proteolysis by thrombin or ADAM12, in which IGF1 is retained, but IGFBP3 CBP3 is released. This intermediate ternary complex can be explained by the previous results showing that CBPs have lower affinities than NBPs for IGF135,36.

The results obtained after proteolysis by thrombin and ADAM12 could not be generalized to other proteases, such as PAPP-A2. The authors should remark this aspect.

Pages 19-20

Given the crucial role of the IGF axis in growth and proliferation and the genetic defects in IGF, IGFbps, and ALS associated with various human diseases, our study also can pave the way for identifying eventual therapeutic targets of IGF axis dysfunction.

Since no genetic defect has yet been reported in any of the six IGFbps, the IGFbps should be omitted for the previous paragraph.

Authors' response to the reviewers' comments
“Structural basis for assembly and disassembly of the IGF/IGFBP/ALS ternary complex” by Kim and Fu et al.

We are grateful to the reviewers for their careful evaluation of our manuscript and for constructive suggestions. In addition to revising the text, we have performed a series of experiments to address the reviewers' concerns. These new results have been incorporated into the revised manuscript. In the text below, the reviewers' comments are in italics and our responses and descriptions of the changes made in the manuscript are in bold blue typeface.

Reviewer #1 (Remarks to the Author):

IGFBP3 or IGFBP5 together with ALS protect the degradation of IGF1 in circulation through forming a stable 1:1:1 ternary complex. Kim et. al here presents the cryo-EM structure of IGF1/IGFBP3/ALS ternary complex, which is a long-awaited structure in IGF field. This structure shows how IGF1, IGFBP3 and ALS are assembled in good detail. Combined with a large body of biochemical experiments, the authors also illustrate the inhibitory role of the CLD domain of IGFBP3 in the complex formation, reveals the process of sequential assembly of the ternary complex, and provides structural explanation for why IGF1/IGFBP3 and IGF1/IGFBP5 but not other IGF1/IGFBPs can interact with ALS. Overall, this is a nice piece of work, and the structural information present here is certainly valuable. However, there are a few major issues need to be addressed before this work could be published. This includes: (1) the resolution for the cryo-EM map is overestimated to some extent. (2) The structural model is not validated by mutagenesis and biochemical experiments. (3) The proposed model that IGF1/IGFBP3-NBP/ALS could activate IGF1R without dissociation is very speculative due to the lack of strong structural and biochemical evidences. Substantial amount of work is required for the revision.

We appreciate the reviewer's positive assessment of our findings as “a long-awaited structure in IGF field” and “a nice piece of work,” and saying that “the structural information present here is certainly valuable.” We hope that our revisions will assuage any remaining concerns.

My specific points are:

(1) The overall resolution for the cryo-EM map is clearly overestimated. The resolution should be calculated based on the FSC between map and model at 0.5 (not 0.143). Based on the model/map FSC curve shown in Extended Figure 1d, the resolution is close to 4 Å, rather than 3 Å that is reported in the paper based map/map FSC. The inconsistent resolution estimation result between using different criteria is due to the over-fitting. The authors should optimize the 3D refinement to prevent the overfitting. The author could try to apply different programs (cryoSPARC versus RELION) to refine the structure.

We appreciate the reviewer's valuable suggestion. To avoid overfitting during 3D refinement, we re-processed the cryo-EM structure of the IGF1/IGFBP3/ALS ternary complex with one selected 3D model among ten 3D classes using cryoSPARC. In brief, we performed non-uniform refinement on the resulting volume using per-particle defocus and per-group CTF optimizations applied. We also re-calculated FSC between new map and new model at 0.5. The resulting resolution are now 3.6 Å (No mask) and 3.2 Å (with tight mask) according to FSC between half maps at 0.143, and 3.6 Å according to the FSC between map and model at 0.5, both of which now show the consistent resolution. We have revised the manuscript accordingly (p. 26 highlighted in yellow, Method section for image processing; Supplementary Fig. 1 and Supplementary Table.1) and attached new validation report that are produced by wwPDB with new map and model.

Supplementary Fig. 1 | Cryo-EM analysis of IGF1/IGFBP3/ALS ternary complex.

(2) Related to my last point. Since the resolution for the cryo-EM map is not very high, the reliability of the model building becomes questionable, especially for ALS whose structure is not reported before. To convince the readers that the model is accurate, the authors need to explicitly show the density of each domain of IGF1/IGFBP3/ALS with the model fitted. It is also important to show the densities for glycosylation and disulfide bonds of ALS. This is a good way to validate the structure. Some key regions of ALS were resolved at even lower resolution, e.g., the hook loop of ALS. The authors need to elaborate in the methods section about how the model building is performed with the poor cryo-EM densities.

As mentioned above, we re-processed the cryo-EM structure and obtained 3.2 Å resolution cryo-EM map (with tight mask). ALS contains 19 LRR repeats and lots of bulky side chain. Moreover, the resolution of central core part of ternary complex is even better than 3.2 Å, as shown in local resolution map (Supplementary Fig. 1e). Therefore, the density map is good enough to build most of side chains and glycans. To address reviewer's comment and convince the reader, we have added the density of each domain and interaction interfaces of IGF1/IGFBP3/ALS ternary complex, and glycans attached to ALS with the model fitted (Supplementary Fig. 2 and Supplementary Fig. 6a).

Supplementary Fig. 2 and Supplementary Fig. 6a | Cryo-EM map for IGF1/IGFBP3/ALS complex and model

d

Supplementary Fig. 2d | Cryo-EM map and model of ALS hook loop (residues 558-579, contour level = 8.5), IGFBP3 C-terminal loop (residues 245-258, contour level = 8.5), and IGF1 C domain (residues 24-43, contour level = 8.5).

However, some regions such as the hook loop and LRRNT of ALS were resolved at lower local resolution ($>3.6\text{\AA}$), as the reviewer mentioned (Supplementary Fig. 2d). Therefore, model building of this ambiguous parts was aided by a protein model generated from AlphaFold2 and a post-processed map generated from DeepEMhancer. This procedure for model building has been added in the method section of revised manuscript (p.26 highlighted in yellow, Method section for model building). Due to the lack of side chain densities at ALS hook loop tip, we also tone down the descriptions about the detail atomic interactions between ALS hook loop tip and IGFBP3 as follows :

“The hook loop of ALS was intercalated between NBP3 and CBP3 to tighten the ALS and IGFBP3 interactions. Although we cannot model the residues of the ALS hook loop tip unambiguously due to the lack of side chain densities, the backbone position of ALS hook loop can be determined by clear cryo-EM density (Supplementary Fig. 2d). Therefore, it is likely that the negative charged residues (E567 and D569) at the ALS hook loop tip contribute to the interaction with IGFBP3’s positive charged residues [R19 (NBP), R206 and R225 (CBP3)] (Fig. 3g and Supplementary Fig. 4e).” (p. 11, highlighted in yellow).

(3) The author didn’t perform any mutagenesis and binding experiment to validate the structural model of 1:1:1 IGF1/IGFBP3/ALS ternary complex. It is, no doubt, that such validation is required when the complex structure is determined for the first time. Therefore, the authors need to introduce some mutations to each interface between IGF1 and ALS and between IGFBP3 and ALS, and test the effect these mutation on the formation of ternary complex by using the pull-down experiments. The interface between IGF1 and IGFBP3 doesn’t need to be validated as similar structure has been determined before.

We appreciate the reviewer’s insightful suggestion. To investigate the significance of these interacting interfaces for ternary complex formation [ALS LRR3-6 motif/IGF1 C-domain loop (IGF1 patch), ALS LRR 5-12 motif/CBP3 α 3 helix (CBP3 helix patch), ALS LRR 13-LRRCT and CBP3 C-terminal loop (CBP3 loop patch), and ALS hook loop/CBP3 loops/NBP3 α 1 helix (hook loop patch)], we selected key interacting residues and introduced mutations (ALS W173A and IGF1 R36E/R37E for the IGF1 patch; IGFBP3 R188E for the CBP3 helix patch; ALS R414A for the CBP3 loop patch; ALS Δ 561-578 deletion for the hook loop

patch). Then, we co-expressed the different combinations of IGF1-Strep-tag, IGFBP3, and ALS-His as indicated in Supplementary Fig. 6b (table) using Expi 293F cells and monitored the components in the resulting complex by pull-down of IGF1 with Strep-Tactin resin.

Supplementary Fig. 6 | Effects on ternary complex formation by mutations of key residues at the interface between ALS and the binary complex

a, Cryo-EM map and model of ALS and CBP3 in CBP3 helix patch (left, contour level = 8.5) and CBP3 loop patch (right, contour level = 9). Key residues for mutation and interacting residues are labeled.

b, SDS-PAGE for pull-down experiments with Expi 293F culture media co-expressing IGF1, IGFBP3, and ALS WT or indicated mutants (left) and quantification of ALS bound to these binary complexes (right, bottom). The mutated key interacting residues at each patch and the construct combination for co-expression are indicated in the table (right, top). The components in the resulting complex were monitored by Coomassie staining after pull-down of IGF1 with Strep-Tactin resin. Expression of ALS mutants-His (W173A, R414A, and hook loop deletion) was confirmed by pull-down with Ni-NTA resin. Densitometric analyses of ALS/IGFBP3 ratios are shown (right, bottom). Data from four independent experiments ($n = 4$) were analyzed and expressed as mean \pm SD (**** $P < 0.0001$ vs. control). P values by one-way ANOVA test followed by Sidak's multiple comparisons test.

Although all mutants were well expressed and the binary complex (IGF1 R36E/R37E-IGFBP3 WT or IGF1 WT-IGFBP3 R188E) was properly assembled, the amount of ALS bound to the binary complex was

significantly reduced, implying that these residues in each interacting patch are indeed critical for ternary complex formation. Particularly, mutations in the CBP3 helix patch and CBP3 loop patch (R188E of IGFBP3 for the CBP3 helix patch and ALS R414A for the CBP3 loop patch) completely abolished ALS binding to the binary complex, indicating that interactions between CBP3 and the ALS concave surface are especially crucial for ternary complex formation and partially explaining why IGF1 alone could not bind to ALS. We have added these new data to the revised manuscript (p. 11, highlighted in yellow, and Supplementary Fig. 6).

(4) The authors proposed that, after the proteolysis, CBP of IGFBP3 would fall apart from the ternary complex and the remaining IGF1/IGFBP3-NBP/ALS could activate IGF1R without further dissociation. Such claim is too speculative, because it is only supported by a cell-based auto-phosphorylation experiment. As IGF1 is localized in the inner surface of half-moon shaped ALS, it is hard to imagine how IGF1/IGFBP3-NBP/ALS could bind the L1/ α CT of IGF1R (IGF1 binding site in IGF1R) without causing any clash between ALS and IGF1R. In addition, previous structure has shown that IGF1 is sandwiched between two protomers in the active state of IGF1R. It is impossible that IGF1/IGFBP3-NBP/ALS could stabilize the active state of IGF1R in a similar fashion as IGF1 does.

Here is my alternative explanation for the functional experiment result. After CBP of IGFBP3 falls apart from ternary complex by proteolysis, the remaining IGF1/IGFBP3-NBP/ALS complex will become relatively instable. It is therefore very likely that IGF1 binds to IGFBP3-NBP/ALS complex in a much lower affinity than to IGF1R (K_d : ~ 1 nM). As a result, the presence of IGF1R would shift the association-dissociation equilibrium of IGF1/IGFBP3-NBP/ALS complex, thereby promoting the quick dissociation of IGF1/IGFBP3-NBP/ALS complex. The released IGF1 from ternary complex is able to activate IGF1R on its own. If the authors want to make a strong claim that IGF1/IGFBP3-NBP/ALS as a whole can bind and activate IGF1R, they need to provide more structural and biochemical evidences. For instance, they could solve the cryo-EM structure of IGF1/IGFBP3-NBP/ALS in complex with IGF1R to confirm that IGF1/IGFBP3-NBP/ALS could bind IGF1R altogether. Such structure would be informative even at low resolution. They could also perform rigorous in vitro binding experiment to show that IGF1/IGFBP3-NBP/ALS could bind IGF1R without the requirement of dissociation.

Reviewer 1 raises a very important point. To address this concern, we purified the IGF1R ecto domain-leucine zipper [PMID: 32459985] and tested whether the resulting intermediate complex after thrombin proteolysis (IGF1/IGFBP3-NBP/ALS) could bind the IGF1R dimer (Supplementary Fig. 7f). We found that the intermediate ternary complex lacking CBP3 after thrombin proteolysis could not directly bind to the purified IGF1R ecto domain dimer, whereas IGF1 alone could do so. Moreover, the presence of the IGF1R ecto domain alone could not promote quick dissociation of IGF1 from this intermediate ternary complex. These results indicate that after the proteolysis of IGFBP3 CLD and CBP3 release, IGF1 needs to be further dissociated from the destabilized IGF1/IGFBP3-NBP/ALS complex for IGF1R activation, which may be mediated by other unknown factors at the extracellular matrix or plasma membrane.

Supplementary Fig. 7 | f, Pull-down of the IGF1R ectodomain-leucine zipper-Strep (IGF1R-zip-strep) with Strep-Tactin resin after mixing IGF1R-zip-strep with IGF1-His or the intermediate ternary complex (IGF1-His/IGFBP3/ALS ternary complex after thrombin proteolysis). Immunoblot analysis was performed with anti-His and anti-Strep antibody to detect IGF1 bound to IGF1R-zip-strep and IGF1R-zip-strep bound to Strep-Tactin resin, respectively (left). Immunoblot analysis was also performed with anti-ALS, anti-IGFBP3, anti-His antibody to detect ALS, IGFBP3, and IGF-His which are bound to IGF1R-zip-strep or are contained in flow through (right).

We have added these new data to the revised manuscript and have revised the subheading of result section and the discussion as follows :

Results subheading: “IGFBP3 CLD proteolysis by thrombin and ADAM12 induces IGFBP3 CBP release and produces the intermediate ternary complex” (p. 15, highlighted in yellow).

“Interestingly, the intermediate ternary complex lacking CBP3 after thrombin proteolysis could not directly bind to the purified IGF1R ecto domain dimer (IGF1R ectodomain-leucine zipper)³⁸, whereas IGF1 alone could do so (Supplementary Fig. 7f). Moreover, the presence of the IGF1R ecto domain alone could not promote the quick dissociation of IGF1 from this intermediate ternary complex (Supplementary Fig. 7f). These results suggest that after the proteolysis of IGFBP3 CLD and CBP3 release, IGF1 needs to be further dissociated from the destabilized IGF1/IGFBP3-NBP/ALS complex for IGF1R activation, which can be mediated by some other unknown factors at the extracellular matrix (ECM) or plasma membrane.” (p. 17, highlighted in yellow and Supplementary Fig. 7f).

“Of greater interest, the amount of released CBP3-GFP from the ternary complex containing hybrid-BPs after thrombin digestion was highly correlated with their ability for IGF1R phosphorylation (hybrid-BP5 >> hybrid-BP2 and -4 > hybrid-BP1 and -6) (Fig. 6f, h), suggesting that the release of CBP3 from the ternary complex after CLD proteolysis destabilizes the intermediate complex and further enhances IGF1 bioavailability.” (p. 17, highlighted in yellow and Supplementary Fig. 7f).

“However, our results suggest that proteolysis of the IGFBP3 CLD is required for IGF1R activation and that after the IGFBP3 CBP falls apart from the ternary complex by proteolysis (by thrombin or ADAM12), the remaining IGF1/IGFBP3-NBP/ALS complex will become relatively unstable, thereby further enhancing IGF1 bioavailability. We also postulate that the potential conformational changes of IGFBP3 CLD after proteolysis by thrombin or ADAM12 can cause subsequent dissociation of CBP from the ternary complex (Fig. 7c, bottom).” (p. 22, highlighted in yellow).

(5) Related to my last point. In Fig.6a, small amount of IGF1 is certainly released from ternary complex after proteolysis. How long does the authors incubate IGF1-GFP/IGFBP3/ALS with thrombin in this experiment? The authors need to perform a time course experiment (like that in Fig.6f) to make sure that the amount of free IGF1 doesn't increase with longer proteolysis time.

In response to the reviewer's valuable suggestions, we have included the protease digestion time in two figures (Fig. 6 and Supplementary Fig. 7) and in a figure legend (p. 41-42, highlighted in yellow). Moreover, we performed a time-course experiment (IGF1-GFP/IGFBP3/ALS + thrombin 0, 10, 16 h), confirming that the amount of free IGF1-GFP did not increase much with longer proteolysis time. We have added these new data to the revised manuscript (p. 15, highlighted in yellow, and Supplementary Fig. 7a).

Supplementary Fig. 7 | a, FSEC of the ternary complex (IGF1-GFP/IGFBP3/ALS) after thrombin digestion for the indicated time (10 and 16 h). Fluorescence signal (GFP) was monitored to examine the dissociation of IGF1 from the ternary complex.

To test whether thrombin digestion of IGF1-GFP/IGFBP3/ALS for 10–16 h can lead to removal of GFP from IGF1, we incubated IGF1-GFP with thrombin for 16 h and checked the results using FSEC. The result showed that the peak position of IGF1-GFP after thrombin digestion is not changed when it compared to that of IGF1-GFP, indicating that thrombin could not cleave IGF1, GFP, or the linker between IGF1 and GFP. We present these data here only for the reviewer's consideration.

Fig. 1 for reviewer only | a, Purification of IGF1-GFP. SEC profile and fluorescence signal of IGF1-GFP. **b**, FSEC of IGF1-GFP with thrombin digestion for 16 h (red) and without thrombin digestion (black).

Reviewer #2 (Remarks to the Author):

This manuscript reports a 3 Å resolution cryo EM structure of the IGF1:IGFBP-3:acid-labile subunit complex. Whilst the structure of ALS has been predicted for some time through homology modelling, this is the first structure of ALS and the first structure of the ternary complex. The ternary complex structure reveals the surfaces of interaction between the three components and confirms previous predictions. Molecular detail of specific sites of interaction between ALS and IGFBP-3 and ALS and IGF1 are identified. The structural information is used to explain effects of clinically relevant mutations. Through a series of pulldown and size exclusion chromatography experiments with the ternary complex and differently tagged components some interesting observations regarding the assembly and proteolytic release of IGF1 are made. Firstly, the data confirm previous experiments that showed the ternary IGF1:IGFBP-3:ALS complex requires prior formation of the binary IGF1:IGFBP-3 complex. Interestingly if the linker-domain is replaced by a linker-domain from either IGFBP-1, -2, -4 or -6 a complex between the hybrid receptors and ALS can be formed, implying that the linker-domains of IGFBP-3 and -5 have unique features that prevent binding unless pre bound to IGF1. Somewhat controversially the authors also suggest that the proteolysis of the IGFBP3 linker domain in the ternary complex induces a linker-domain conformational change and subsequent dissociation of BP3 C-domain from the ternary complex. The suggestion is that a IGF1 remains in the high molecular weight complex and a hydrophobic patch on IGF1 is exposed that is critical for IGF1R binding, rather than the current understanding that proteolysis results in release of free IGF1 that then binds and activates the IGF-1R. To definitively show this it would be good to isolate the digested complex from CBP-3 and test whether this is able to activate the IGF-1R. Is it possible to predict whether the entire complex with the exposed IGF1 hydrophobic patch could be accommodated in the IGF-1R binding site as suggested? This manuscript represents a significant body of data that provides insight into the mechanism of action of the ternary IGF1:IGFBP-3:ALS complex.

We appreciate the reviewer's assessment of our work as "a significant body of data that provides insight into the mechanism of action of the ternary IGF1 : IGFBP3 : ALS complex."

Specific comments/corrections

Page 2 first sentence "The structure of the IGF1/IGF1R complex also explains why, as compared with IGF1, IGF2 and insulin with short C-domain loops exhibit weaker binding to IGF1R" This sentence is inaccurate or ambiguous. The mature insulin does not have a C-domain. Also it would be better to say that the structure of the IGF1/IGF-1R complex reveals why the IGFs are unable to bind the IGF-1R if they exist in an IGF:IGFBP binary complex. The binding pocket is too small to accommodate the binary complex and the IGF residues involved in the IGF-1R primary high affinity binding site are engaged in the interaction with the IGFBP, thereby preventing binding to the receptor.

In response to the reviewer's valuable suggestions, we have revised this part as follows (p. 4, highlighted in yellow):

"The structure of the IGF1/IGF1R complex explains how only one IGF1 molecule binds the Γ-shaped asymmetric IGF1R dimer for IGF1R activation and why IGF1 with longer C-domain loops than IGF2 exhibits stronger binding to IGF1R¹³. Moreover, these structures also suggest that the IGF1 binding

pocket in IGF1R is too small to accommodate the binary complex and that the IGF residues involved in high-affinity binding to the IGF1R primary site are engaged in interaction with IGFBP, thereby preventing binding of the IGF/IGFBP binary complex to the receptor.”

Page 4 last paragraph “the IGF1/IGFBP3 binary engages” should be “the IGF1/IGFBP3 binary complex engages”

We appreciate the reviewer’s careful inspection of our manuscript. We have revised this phrase accordingly (p. 5, highlighted in yellow).

Page 5 end of discussion. It is not evident what the “intermediate ternary complex that is crucial for IGF1R activation” is. It has not been defined until the discussion.

We appreciate the reviewer’s helpful suggestion. We have revised as follows (p. 5, highlighted in yellow):

“We also found that the CLD of IGFBP3 sterically blocks ALS binding unless IGFBP3 binds to IGF1, and that proteolysis of IGFBP3 CLD in the ternary complex induces release of CBP3 rather than IGF1 from the ternary complex.”

Page 7 second sentence. “nine highly conserved disulfide bonds (3 in IGF1, 6 in NBP3, and 3 in CBP3) crucial for rigidity of the core modules.” There are 12 disulfide bonds described here in brackets. Which 9 are you referring to?

We apologize for the inadequate description of our original data. There are three disulfide bonds in IGF1 and nine disulfide bonds in IGFBP3 (6 in NBP3 + 3 in CBP3). We have revised this sentence accordingly (p. 8, highlighted in yellow).

It would be good to define which residues of the N domain form the “thumb” structure as this region is discussed at length in the results.

In response to the reviewer’s helpful suggestion, we have defined the thumb region of IGFBP as follows and added the corresponding reference [PMID: 15642270] (p. 8, highlighted in yellow):

“The thumb region consists of a short stretch of the very first N-terminal residues of IGFBPs that precede the first N-terminal cysteine (amino acids 1–12 in IGFBP3)²¹.”

IGF1 α B helix is not a common terminology – perhaps describe as the IGF1 B domain α helix. Similarly, it is not clear what the IGF1 Nterminal loop (L5) is – please define. The term IGF1 α B wedge helix is not common and needs to be defined. The terminology “CBP3 β 4 and IGF1 β B” is not defined.

We apologize for omitting the definition of each domain. In response to the reviewer’s valuable suggestion, we have defined them more clearly in revised text and figure legend as follows:

“Residues (L5, A13, F16, L54 and L57) on the IGF1 N-terminal loop (G1-L5) and one face of the IGF1 B domain α helix and IGF1 A domain α 2 helix (L54-Y60) constituted long-range hydrophobic networks with residues on NBP3 (P38, C54, I56, C67, L77, L80, and L81) (Fig. 2d)” (p. 8, highlighted in yellow).

“Of note, our cryo-EM structure identified a unique feature of the IGF1/IGFBP3 binary complex in which the NBP3 thumb is sandwiched between CBP3 β 4 (N201-L203) and IGF1 B domain β strand (G22-Y24), forming a well-organized short β -sheet (Fig. 2f, Supplementary Figs. 3 and 4b)” (p. 9, highlighted in yellow).

“The α helix (A8-V17) and β strand (G22-Y24) of the IGF1 B domain and the α 1 helix (I43-F49) and α 2 helix (L54-Y60) of the IGF1 A domain are labeled as α B, β B, α A1, and α A2, respectively” (Fig. 2 legend and Supplementary Fig. 3c legend, p. 40, highlighted in yellow).

End of page 8. Perhaps better worded as “the thumb region of NBPs is of different length (NBP3, 12; NBP5-6aa, NBP1-4aa, NBP2-6aa, NBP4-5aa, NBP6-1aa amino acids long)”.

In response to the reviewer’s comment, we have rephrased this sentence in the revised manuscript as follows (p. 9, highlighted in yellow):

“Although most residues on the surface of the IGFBP3 cleft for IGF1 interaction are predominantly conserved in IGFBPs, the thumb regions of the different NBPs are of different amino acid (aa) lengths (NBP1, 4 aa; NBP2, 6 aa; NBP3, 12 aa; NBP4, 5 aa; NBP5, 6 aa; NBP6, 1 aa).”

It is not clear if the binary complex being described is just the complex within the ternary complex or whether binary complex particles were detected when collecting data for the ternary complex. There is no specific methods section describing generation of the binary complex for cryo EM and the collection of binary complex cryo EM data. No particle data is provided.

In this study, we determined the cryo-EM structure of the ternary complex (IGF1/IGFBP3/ALS). So far, the structure of the binary complex itself (IGF1/IGFBP3) has not been determined. Therefore, we described the structure of the IGF1/IGFBP3 binary complex within the ternary complex in detail. For greater clarity, we have detailed this as follows (p. 8, highlighted in yellow):

“The IGF1/full-length IGFBP3 complex within the ternary complex adopts similar global folds with the human IGF1/IGFBP4 complex (NBP4 and CBP4 fragment without CLD; PDB ID: 2DSR), with a backbone root mean square deviation (RMSD) of 1.189 Å (Fig. 2c).”

Page 11: “the mutation on Leu and Asp residues” should read “the mutation on Leu and Asn residues”

We appreciate the reviewer’s careful inspection of our manuscript. We have revised the phrase accordingly (p. 12, highlighted in yellow).

“Although IGF1 R36Q and V44M are located at the IGF1 interaction interface with ALS and CBP3, respectively, these mutations do not seem to affect IGF1 binding to ALS or IGFBP3 (Fig. 4e).” – authors should refer to “Netchine et al (2009) DOI 10.1210/jc.2009-0452” and “Walenkamp MJ et al (2005) DOI: 10.1210/jc.2004-1254 which demonstrate that R36Q and V44M IGF-I can still bind IGFBP-3, respectively, and V44M is also able to assemble into the ternary complex (Walenkamp et al).

We appreciate the reviewer’s helpful suggestion. We have added these corresponding references [PMID: 15769976 and 19773405] to the revised manuscript (p. 13, highlighted in yellow). Moreover, based on reviewer 3’s suggestion, we have added one more reported IGF1 patient mutation (Y60H) and revised the manuscript as follows:

“IGF1 deficiency and several bio-inactive IGF1 mutations (R36Q, V44M, R50W, and Y60H) also have been identified in patients with growth failure^{26, 27, 28} (Fig. 4d). IGF1 Y60H, which is highly conserved among species and also conserved in IGF2 and insulin, is far away from the interaction interfaces of the ternary complex, whereas IGF1 R36Q and V44M lie near the IGF1 interaction interface with ALS and CBP3, respectively. However, none of these mutations seem to affect IGF1 binding to ALS or IGFBP3 (Fig. 4e). Therefore, their decreased affinity to IGF1R may result in phenotypes such as severe intrauterine and postnatal growth retardation, microcephaly, and sensorineural deafness^{22, 26}.”

Fig. 4 | d, Mutated IGF1 residues in human patients with growth impairment are indicated in a schematic representation (left) and presented as green spheres and labeled in a cartoon structure of human IGF1 (right). **e,** Close-up views of three IGF1 mutants (Y60H, R36Q, and V44M) and their neighboring residues.

Surprisingly, we found that even in the absence of IGF1, ALS could bind to hybrid-BP1,-2, -4, and -6, but not to IGFBP3 or hybrid-BP5

What is the affinity of the hybridBPs for ALS? Rather than a structural change in the linker upon IGF1 being the mechanism by which BP-3 and BP-5 can then bind ALS, could this be directly due to differences in amino acid sequences? There are a lot of basic residues in BP-3 and BP-5 linker domains.

In response to the reviewer’s suggestion, we compared the amino acid composition and length of the IGFBP CLD, but we could not find any meaningful features that are present only in IGFBP3 and IGFBP5.

	Sequence	aa	Acidic (%)	Basic (%)	Hydrophobic (%)
CLD1	SDASAPHAAFAGSPESPESTEITEEELLDNFHLMAPSEEDHSILWDAISTYDGSKALHVTNIKKW	65	14 (22%)	7 (11%)	27 (42%)
CLD2	RDAEYGASPEQVADNGDDHSEGLVENVDSIMNMLGGGGSAGRKPLKSGMKE LAVFREKVTIEQHRQMGKGGKHHLLGLEEPKKLRPPPA	89	14 (16%)	18 (20%)	30 (34%)
CLD3	SAVSRLLRAYLLPAPPAPGNASESEEDRSAGSVESPSVSSSTRVSDPKFPHLHSKIILIKKGHAKDSQRYKVDYESQSTDTQNFSESKRETE	92	13 (14%)	17 (18%)	32 (35%)
CLD4	AETEAIQESLQPSDKDEGDHPNNSFSPCSAHDRRCLQKHFAKIRDRSTSGGKMKVNGAPREDARPVP	67	11 (16%)	14 (21%)	22 (33%)
CLD5	KSYREQVKIERDSREHEEPTTSEMAEETYSPKIFRPKHTRISELKAFAVKKDRRKKLTQSKFVGGAEHTAHPRIISAPEMRQESE	85	16 (19%)	22 (26%)	27 (32%)
CLD6	RAPAVAEEENPKESKPOAGTARPODVNRRDQQRNPGTSTTPSQPNSAGVQDTE	52	7 (13%)	7 (13%)	16 (31%)

Fig. 1 for reviewer only | Sequence composition analysis of IGFBPs CLD.

However, we do not rule out the possibility that differences in amino acid composition and length of the IGFBP CLD may affect binding of hybrid-BP1, 2, 4, and 6 to ALS even without IGF1. Therefore, we have addressed this issue in the discussion section of the revised manuscript as follows (p. 22, highlighted in yellow):

“However, we do not exclude the possibility that the unique features such as amino acid composition and length of the IGFBP3 CLD and/or the conformational changes in the CBP3 loop patch of IGFBP3 upon IGF1 binding are also involved in sequential assembly of the ternary complex.”

We hope the reviewer understands and agrees that clarifying these issues is beyond the scope of our current study. However, we do expect to address these issues in a follow-up study by determining the Cryo-EM structure of ALS/hybrid BPs (hybrid-BP1, 2, 4, or 6) or by monitoring the dynamics of IGFBP CLD in the binary and ternary complexes with high-speed atomic force microscopy.

Fig 6 legend. It is not clear how long the thrombin cleavage was conducted for in fig a, g and h. Please include this in the legend.

In response to the reviewer’s comment, we have included the protease digestion time in two figures (Fig. 6 and Supplementary Fig. 7) and a figure legend (p. 41-42, highlighted in yellow).

Is there a possibility that in the time frame of the thrombin cleave in fig 6g that there could be some proteolysis leading to removal of the GFP from IGF1?

IGF1-GFP was released from the binary complex after 10 h thrombin digestion and eluted at the peak position corresponding to free IGF1-GFP (Fig. 6a) rather than GFP peak position. Therefore, it is unlikely that thrombin proteolysis leads to removal of the GFP from IGF1. To test this possibility, we incubated IGF1-GFP with thrombin for 16 h and checked the results using FSEC. The result showed that the peak position of IGF1-GFP after thrombin digestion is not changed when it compared to that of IGF1-GFP, indicating that thrombin could not cleave IGF1, GFP, or the linker between IGF1 and GFP. We present these data here only for the reviewer's consideration.

Fig. 2 for reviewer only | a, Purification of IGF1-GFP. SEC profile and fluorescence signal of IGF1-GFP. **b**, FSEC of IGF1-GFP with thrombin digestion for 16 h (red) and without thrombin digestion (black).

Reviewer #3 (Remarks to the Author):

The MS entitled “Structural basis for assembly and disassembly of the IGF/IGFBP/ALS ternary complex” by Hyojin Kim, Yaoyao Fu, Ho Jeong Hong, Seong-Gyu Lee, Dong Sun Lee, and Ho Min Kim from the School of Medical Science and Engineering, and the Advanced Institute of Science and Technology (KAIST), both from the Republic of Korea is a very interesting study aiming to understand the structural complexity of the IGF/IGFBP/ALS ternary complex.

Interestingly, this is the first report of the spatial structure of ALS by using Cryo-EM to overcome the difficulties to obtain a crystalized ALS protein suitable for X-ray diffraction studies. In addition, the spatial structure for the entire IGF/IGFBP/ALS ternary complex was also resolved by image processing with an acceptable overall resolution of 3.0 Å.

The authors also characterize the crucial determinants for the sequential assembly of IGF1 with IGFBP3, and then the binary complex with ALS. Finally, a series of elegant in vitro experiments show that in the absence of IGF1, the central linker domain (CLD) of IGFB3 blocks the interaction with ALS and that the proteolysis of CLD (at least when thrombin was the selected protease) is crucial for the availability of IGF1 for interaction with its receptor. The MS is well written, and the findings are clearly presented, however, there are several points that should be clarified.

We appreciate the reviewer’s overall positive assessment of our paper and hope that our revisions will assuage any remaining concerns.

Abstract

“ . . . suggesting a structural basis for human diseases associated with IGF, IGFBP3, and ALS, such as complete ALS deficiency (ACLSL) and Laron syndrome.”

The associated human diseases the authors want to refer to are the complete ALS deficiency (ACLSL; OMIM #615961) and IGF1 deficiency (OMIM #608747). The Laron syndrome is caused by inactivating mutations/deletions in the GH receptor and the associated IGF1 deficiency results from the impairment of GH action. Indicating the OMIM numbers would be useful for the readers to access to the related human diseases that result from genetic defects in the genes coding for the members of the IGF/IGFBP/ALS ternary complex, mainly IGF1 and IGFALS, since so far, no human mutation has been reported in the IGFBP3 gene.

I would like to suggest changing the paragraph to read as follow:

. . . suggesting a structural basis for human diseases associated with IGF1 and IGFALS gene mutations, such as complete ALS deficiency (ACLSL, OMIM #615961) and IGF1 deficiency (OMIM #608747).

We appreciate the reviewer’s insightful suggestion. We have revised the Abstract and Introduction accordingly (p. 2, p. 4, and p. 5, highlighted in yellow). Because of the format for the abstract, the OMIM

numbers for ALS deficiency (ACLS, OMIM #615961) and IGF1 deficiency (OMIM #608747) have been added in the introduction section.

Main

Page 3

“IGFs can form binary complexes (10%–15%) with IGF-binding proteins (IGFBPs: IGFBP1, IGFBP2, IGFBP4, or IGFBP6), or ternary complexes (80%–90%) with IGFBPs (IGFBP3 or IGFBP5) and acid-labile subunit (ALS)²⁻⁴. In the circulation, free IGFs have a half-life of less than 10 min, but a binary complex increases this half-life to 30–90 min, whereas a ternary complex maintains it up to 16–24 h⁵.”

Change to read as follow:

In the circulation, IGFs can form binary complexes (10%–15%) with IGF-binding proteins (IGFBPs: IGFBP1, IGFBP2, IGFBP4, or IGFBP6), or ternary complexes (80%–90%) with IGFBPs (IGFBP3 or IGFBP5) and acid-labile subunit (ALS)²⁻⁴. In the circulation, Free IGFs have a half-life of less than 10 min, but a binary complex increases this half-life to 30–90 min, whereas a ternary complex maintains it up to 16-24h.

In response to the reviewer’s comment, we have changed the text as follows (p. 3, highlighted in yellow):

“In the circulation, IGFs can form binary complexes (10%–15%) with IGF-binding proteins (IGFBPs: IGFBP1, IGFBP2, IGFBP4, or IGFBP6), or ternary complexes (80%–90%) with IGFBPs (IGFBP3 or IGFBP5) and acid-labile subunit (ALS)²⁻⁴. Free IGFs in serum have a half-life of less than 10 min, but a binary complex increases this half-life to 30–90 min, whereas a ternary complex maintains it up to 16–24 h⁵.”

Page 4

“Individuals with complete ALS deficiency (ACLS) have mutations in the IGFALS gene and exhibit severely reduced serum IGF1 and IGFBPs (particularly IGFBP3), leading to low birth weight and length, reduced head circumference and height, pubertal delay, and insulin resistance^{15,16}.”

I would like to suggest adding the reference of the original description of ALS deficiency and again the OMIM#. ACLS: OMIM #615961. Domené HM, N Engl J Med 350, 570-7 (2004).

As suggested, we have cited the corresponding reference [PMID: 14762184] and the OMIM number [OMIM #615961] (p. 4, highlighted in yellow).

Page 5

“Taken together, our findings provide mechanistic insight into assembly of the IGF/IGFBP3/ALS ternary complex and its disassembly for IGF bioavailability and allowed us to interpret the structural effects of IGF1 and ALS mutations in ACLSD and Laron syndrome.”

Change to:

Taken together, our findings provide mechanistic insight into assembly of the IGF/IGFBP3/ALS ternary complex and its disassembly for IGF bioavailability and allowed us to interpret the structural effects of IGF1 and ALS mutations in ACLSD and IGF1 deficiencies.

We appreciate the reviewer’s insightful suggestion. We have revised the Introduction accordingly and added OMIM numbers for ACLSD [OMIM #615961] and IGF1 deficiency [OMIM #608747] (p. 4 and p. 5, highlighted in yellow).

Page 8

It has been reported that the extended N-terminal thumb of NBP is critical not only for the regulation of IGF1 binding to IGF1R but also for interactions between NBP and CBP.

The authors should add a reference to this assessment.

We apologize for missing this reference. In response to the reviewer’s comment, we have cited the corresponding reference [PMID: 16924115] (p. 8, highlighted in yellow).

Page 11, Structural basis for disease-associated ALS and IGF1 mutations

“To date, various mutations (18 missense, 7 frameshift, 2 in-frame insertion, 1 nonsense mutation, and 1 deletion of exon 2) in the human ALS gene have been reported in patients with ACLSD16 (Fig. 4a, b). Although the 18 missense mutations are located in the LRRNT (C60S and P73L), LRR (N84S, L127P, L134Q, T145K, L172F, L213F, L241P, L244F, N276S, L409F, D440N, A475V, D488N, and S490W), and LRRCT domains (C540R and P586L), none of the residues are involved in direct interactions with the IGF1/IGFBP3 binary complex.”

As it was reported in reference #16 (Domene S & Domene HM, Mol Cell Endocrinol 518, 111006, 2020) 28 different mutations have been described in ACLSD (17 missense, 7 frameshift, 2 in-frame insertion, 1 nonsense, and 1 deletion of the entire exon 2). In some patients two variants have been found in cis (L409F and A475V). In vitro expression of these variants and functional ternary complex analysis demonstrates that while the A475V variant is benign, is able to be expressed, secreted and retains its ability for ternary complex formation, the L409F variant is not expressed in vitro and is the one responsible for the complete ACLSD of the patient (Martucci LC, et al. Mol Cell Endocrinol 429, 19-28, 2016).

The A475V should be eliminated from the text, to read as follow:

To date, various mutations (17 missense, 7 frameshift, 2 in-frame insertion, 1 nonsense mutation, and 1 deletion of exon 2) in the human ALS gene have been reported in patients with ACLSD16 (Fig. 4a, b). Although the 18 missense mutations are located in the LRRNT (C60S and P73L), LRR (N84S, L127P, L134Q, T145K, L172F, L213F, L241P, L244F, N276S, L409F, D440N, A475V, D488N, and S490W), and LRRCT domains (C540R and P586L), none of residues are involved in direct interactions with the IGF1/IGFBP3 binary complex.

In addition, the reference from Martucci et al. should be added.

We appreciate the reviewer's careful inspection of our manuscript. We have revised the phrasing accordingly and cited the corresponding reference [PMID: 27018247] (p. 12, highlighted in yellow).

Figure 4

Figure 4 presents several inconsistencies. First, to the knowledge of this reviewer the variants C60F and the A434V have not been previously described. As was previously mentioned, the following variants: R277H, P287L, A475V, and R548W have been described in children with short stature but not associated to complete ACLSD (Domené HM et al. *Horm Res Paediatr* 80, 413-23, 2013). In addition, similarly to the A475V variant, P287L and R548W are normally expressed and secreted in vitro and retain their ability to form ternary complexes.

For all these reasons, variants C60F, R277H, P287L, A434V, A475V, and R548W, should be removed from both the schematic representations of Figure 4a and 4b (Martucci LC, et al. *Mol Cell Endocrinol* 429, 19-28, 2016).

We appreciate the reviewer's valuable comment. In the revised figure panels (4a and 4b), we have removed variants C60F, R277H, P287L, A434V, A475V, and R548W, which are not associated with complete ACLSD.

Fig. 4 | a, Schematic representation of the ALS protein indicating the location of the 28 identified mutations from patients with ASCLD. **b**, Mutated residues in patients with ASCLD (only point mutations) are presented as green spheres and labeled in the cartoon structure of human ALS.

Pages 11-12

“In particular, the mutation on Leu and Asp residues (N84S, L127P, L134Q, L172F, L241P, L244F, N76S, and L409F) as part of the LRR consensus sequence (xLxxLxLxxNxLxxLxxxxFxx/Lx) are likely to introduce a main chain distortion or disrupt the LRR hydrophobic core, affecting the entire LRR architecture of the ALS protein. Similar to this effect, substitution of T145 and L213 residues to long charged Lys and bulky hydrophobic Phe, respectively, which point toward the hydrophobic core of the LRR solenoid interior, would introduce steric hindrance with surrounding residues and destroy the regular fold of the LRR module. The highly conserved C60 and C540 residues participate in disulfide bridge formation with C45 and C583, respectively, and the P73 and P586 residues contribute to maintaining structural rigidity. Therefore, mutations at these residues (C60S, C540R, P73L, and P586L) could destabilize LRRNT and LRRCT (Fig. 4c). Although D440 and D448 are not essential to ALS/IGFBP3 interactions, D440N and possibly D488N could introduce a new N-glycosylation site that leads to impairment of secretion of ALS and its ability to form a ternary complex, as reported previously²². The S490W mutation is predicted to cause loss of the hydrogen bond network with neighboring S466, R467, D488, H491, S512, and R514 on the surface of the LRR17-LRR19 ascending flank (Fig. 4c). Two in-frame mutants, S195_R197dup and L437_L439dup, insert three amino acids at LRR6 and LRR16, respectively. The resulting altered length of the LRR modules might impair alignment of the hydrophobic residues on the consensus motif, perturbing the solenoid-like structure of ALS. Moreover, S195_R197dup and L437_L439dup lie adjacent to the ternary complex interface, probably disrupting ternary complex formation”

In the previous paragraph the authors analyze the effect of several IGFALS gene variants on the overall structure of the ALS protein. As they rightly mention, D440N and D488N could introduce a novel N-glycosylation site that may result in secretion impairment (this was already demonstrated by Firth SM et al. reference 22).

The authors should also mention that David et al. (reference 19) already analyzed variants L134Q, L172F, L241P, L244F, N276S, C60S, C540R, P73L, S195_R197dup and L437_L439dup and according to their own predictive ALS spatial structure all of them resulted in major distortions in ALS structure. By using the same structural model, Martucci et al. also reported major distortion in L213F and L409F. In vitro expression of these ALS variants demonstrates that while L409F is not synthesized, L213F is partially expressed but is unable to be secreted remaining trapped inside the cells. These previous finding should be mentioned.

In response to these helpful suggestions, we have revised the text as follows and cited two corresponding references [PMID: 22991227 and 27018247] (p. 12 and p. 13, highlighted in yellow):

“Our ALS cryo-EM structure clearly showed that these mutations can affect the structural integrity of ALS, possibly causing its misfolding and aberrant secretion (Fig. 4c), which are consistent with a previous analysis using the predicted ALS structural model²⁰.”

“Indeed, previous studies on the *in vitro* expression of ALS variants demonstrated that N276S, L409F, and C540R are not synthesized, whereas L213F is partially expressed but cannot be secreted²⁴.”

“IGF1 deficiency and several bio-inactive IGF1 mutations (R36Q, V44M, and R50W) also have been identified in patients with growth failure 23,24 (Fig. 4d). Although IGF1 R36Q and V44M are located at the IGF1 interaction interface with ALS and CBP3, respectively, these mutations do not seem to affect IGF1 binding to ALS or IGFBP3 (Fig. 4e). However, the decreased affinity of these IGF1 mutants for IGF1R may result in phenotypes such as severe intrauterine and postnatal growth retardation, microcephaly, and sensorineural deafness²³. Taken together, our findings provide a structural basis for ACLSD and the impaired growth seen with genetic defects in the human IGF1/IGFBP3/ALS ternary complex.”

The authors omitted one reported IGF1 mutation which resulted in pre- and postnatal growth retardation, microcephaly, facial dysmorphism, bilateral sensorineural deafness, and mild global development delay (Keselman et al. *Europ J Endocrinol* 181, K43-K53, 2019). Tyr60 residue is highly conserved among species, and also conserved in IGF2 and insulin. Although the reported Y60H would not affect the interaction with IGFBP3, Y60 is a key residue involving in IGF1R binding (Bayne ML et al. *J Biol Chem* 265,15648-52, 1990).

These references should also be added, analyzed in the text, and reported in Figure 4d.

We apologize for missing a reported IGF1 mutation (Y60H). As previously suggested, our ternary complex structure showed that IGF1 Y60H is not located at the IGF1 interaction interface with IGFBP3 or ALS. Therefore, we have revised the text and Figure 4d and 4e as follows and cited two corresponding references [PMID: 31539878 and 2168421] (p. 13, highlighted in yellow):

“IGF1 deficiency and several bio-inactive IGF1 mutations (R36Q, V44M, R50W, and Y60H) also have been identified in patients with growth failure^{26, 27, 28} (Fig. 4d). IGF1 Y60H, which is highly conserved among species and also conserved in IGF2 and insulin, is far away from the interaction interfaces of the ternary complex, whereas IGF1 R36Q and V44M lie near the IGF1 interaction interface with ALS and CBP3, respectively. However, none of these mutations seem to affect IGF1 binding to ALS or IGFBP3 (Fig. 4e). Therefore, their decreased affinity to IGF1R may result in phenotypes such as severe intrauterine and postnatal growth retardation, microcephaly, and sensorineural deafness^{22, 26.}”

Fig. 4 | d, Mutated IGF1 residues in human patients with growth impairment are indicated in a schematic representation (left) and presented as green spheres and labeled in a cartoon structure of human IGF1 (right). **e**, Close-up views of three IGF1 mutants (Y60H, R36Q, and V44M) and their neighboring residues.

Page 14, Release of IGFBP3 CBP from ternary complex upon IGFBP3 CLD proteolysis is crucial for IGF1R activation.

“Proteolysis at the IGFBP CLD by proteases such as metalloproteinase, cathepsins, and serine proteases is crucial for activation of IGF1R signaling in a target tissue or cell^{10,25}. To check whether IGFBP proteolysis causes IGF1 release from the ternary complex, we incubated the purified IGF1-GFP/IGFBP3 binary complex or IGF1-GFP/IGFBP3/ALS ternary complex with thrombin, which cleaves R97-A98 and R206-G207 of IGFBP3²⁵, and used FSEC to monitor IGF1-GFP release from the complex. As expected, IGF1-GFP was released from the binary complex after thrombin digestion and eluted at the peak position, corresponding to free IGF1-GFP (Fig. 6a). Strikingly, there was no free IGF1-GFP peak released from the ternary complex after thrombin digestion, in contrast to previous reports²⁶.”

Several proteases are responsible for IGFBP3 (and IGFBP5) proteolysis. They have different proteolytic sites on the CLD3 resulting in different fragments and, potentially in different IGFBP3 fragments. Thrombin cleaves specifically the CLD of IGFBP3 at two sites: R97-A98 and R206-G207 (Booth BA et al. Am J Physiol 271, E465-70, 1996; Firth SM & Baxter RC Endocr Rev 23, 824-54, 2002 (ref.25)). PAPP-A2 (a specific protease for IGFBP3 and IGFBP5) appears to constitute the most relevant physiologic protease present in the circulation. It has been shown that PAPP-A2 cleaves IGFBP5 at S143-K144 but no report indicating the specific digestion site for IGFBP3 has yet been reported. However, it is likely that the fragments resulting from PAPP-A2 proteolysis could have less ability to retain IGF1. This could explain the different results obtained by Lee & Rechler (Ref26) when they incubated ternary complex with adult rat serum and observed release of IGF1 after proteolysis of IGFBP3 that was then transferred to rat IGF1R during the incubation (reference 26 in the MS). The physiological relevance of PAPP-A2 is further emphasized by the description of PAPP-A2 deficiency, as a result of inactivating mutations in the PAPP2 gene, characterized by short stature associated to high levels of IGF1, IGFBP3, IGFBP5 and ALS (Dauber A et al. EMBO Mol Med 8, 363-74, 2016) (PAPP-A2, OMIM# 619489).

We agree with the reviewer’s insightful suggestion. We have now included the description of PAPP-A2 deficiency and added the corresponding reference and OMIM number [PMID: 26902202 and OMIM# 619489] as follows (p. 16, highlighted in yellow). We also performed additional experiments with PAPP-A2 (see below).

“It has been known that a disintegrin and metalloprotease 12 (ADAM12) and pregnancy-associated plasma protein A2 (PAPP-A2) in pregnancy serum are also responsible for proteolysis of IGFBP3 and IGFBP5^{34, 35}, and mutations in PAPP-A2 lead to short stature because of low IGF1 availability (OMIM# 619489)³⁶”.

Page 15

“Of interest, the CBP3 band was not detected after thrombin digestion and subsequent SEC, whereas ALS, IGF1, and cleaved NBP3 remained in the ternary complex (Fig. 6b). These results indicate that CBP3, not IGF1, is released from the ternary complex after proteolysis at the CLD of IGFBP3.”

There is no experimental data supporting that the lack of CBP3 in the complexes obtained after thrombin digestion can be extended to all other IGFBP3 proteases, particularly to PAPP-A2. It is possible that different proteases, having their private digestion sites, may produce different fragments of IGFBP3 retaining or not CPB3.

The authors should clarify this point.

Here, Reviewer 3 raises a very important point. To address the reviewers' comments, we additionally investigated whether the results obtained after proteolysis by thrombin could be generalized to that by ADAM12 and PAPP-A2. Of note, we found that similar to thrombin, ADAM12 efficiently degraded IGFBP3 in the ternary complex, which in turn induced the release of CBP3-GFP, but not IGF-GFP (Supplementary Fig. 7b–7d). However, compared with thrombin and ADAM12, the efficiency of IGFBP3 cleavage by PAPP-A2 was significantly reduced, whereas PAPP-A2 properly cleaved hybrid-BP5 (Supplementary Fig. 7b and 7h). These results are consistent with those of previous reports showing limited degradation of IGFBP3 by PAPP-A2 but complete proteolysis of IGFBP5 by PAPP-A2 [PMID: 11264294, Overgaard MT et al., JBC, 2001]. Of interest, even after PAPP-A2 treatment of the IGF1/IGFBP3/ALS ternary complex or IGF1/hybrid-BP5/ALS ternary complex, neither CBP3-GFP nor IGF-GFP was released (Supplementary Fig. 7c, d, i, and j).

Supplementary Fig. 7 | b, SDS-PAGE analysis of the IGF1/IGFBP3/ALS ternary complex after proteolysis (0.2 units thrombin/1 μ g ternary complex for 10 h; 0.2 μ g ADAM12/1 μ g ternary complex for 16 h; 0.2 μ g PAPP-A2/1 μ g ternary complex for 16 h). **c and d**, FSEC of the ternary complex IGF1-GFP/IGFBP3/ALS (c) or IGF1/IGFBP3-GFP/ALS (d) after

ADAM12 (Cyan, 0.2 μ g ADAM12/1 μ g ternary complex for 16 h) or PAPP-A2 (Green, 0.2 μ g PAPP-A2/1 μ g ternary complex for 16 h) digestion. Fluorescence signal (GFP) was monitored to examine the dissociation of CBP3 or IGF1 from the indicated ternary complex. **h**, SDS-PAGE analysis of the IGF1/hybrid-BP5/ALS ternary complex after proteolysis (0.2 unit thrombin/1 μ g ternary complex for 10 h; 0.2 μ g ADAM12/1 μ g ternary complex for 16 h; 0.2 μ g PAPP-A2/1 μ g ternary complex for 16 h). The IGF1/hybrid-BP5/ALS ternary complex was used for control (Ctrl). **i** and **j**, FSEC of the ternary complex IGF1-GFP/hybrid-BP5/ALS (**i**) or IGF1/hybrid-BP5-GFP/ALS (**j**) after ADAM12 (Cyan, 0.2 μ g ADAM12/1 μ g ternary complex for 16 h) or PAPP-A2 (Green, 0.2 μ g PAPP-A2/1 μ g ternary complex for 16 h) digestion. Fluorescence signal (GFP) was monitored to examine the dissociation of IGF1 (**i**) or CBP3 (**j**) from the indicated ternary complex.

Of note, the original ternary complex itself did not induce IGF1R phosphorylation, but the purified intermediate ternary complex that resulted from proteolysis with thrombin and ADAM12 could activate IGF1R signaling at a level similar to that with free IGF1 (Fig. 6d and Supplementary Fig. 7e). The intermediate ternary complex after PAPP-A2 proteolysis, which retained CBP3, could also activate IGF1R signaling, but with less potency as compared with those of the intermediate ternary complex lacking CPB3 after thrombin and ADAM12 proteolysis (Fig. 6d and Supplementary Fig. 7e).

Supplementary Fig. 7e and Fig. 6d | Immunoblot analysis for IGF1R phosphorylation after treatment of the intermediate ternary complex (Supplementary Fig. 7e, left) and its quantification (Fig. 6d, right). HEK293A cells were treated with IGF1 (5 nM, positive control), IGF1/IGFBP3/ALS ternary complex (5 nM, none), or intermediate ternary complex (5 nM). The intermediate ternary complex was prepared by protease digestion (0.2 unit thrombin/1 μ g ternary complex for 10 h; 0.2 μ g ADAM12/1 μ g ternary complex for 16 h; 0.2 μ g PAPP-A2/1 μ g ternary complex for 16 h) and subsequent SEC purification. No treatment was used for negative control (Ctrl). Data from three independent experiments ($n = 3$) were analyzed, and relative pIGF1R / IGF1R (Fold to Ctrl) were expressed as mean \pm SD (ns, not significant; **** $P < 0.0001$ vs. ctrl; ##### $P < 0.0001$ vs. none). P values by one-way ANOVA test followed by Sidak's multiple comparisons test.

Using a purified IGF1R ecto domain-leucine zipper [PMID: 32459985], we then tested whether the intermediate complex after thrombin proteolysis (IGF1/IGFBP3-NBP/ALS) could bind the IGF1R dimer, (Supplementary Fig. 7f). The intermediate ternary complex lacking CBP3 after thrombin proteolysis

could not directly bind to the purified IGF1R ecto domain dimer, while IGF1 alone could do so. Moreover, the presence of the IGF1R ecto domain alone could not promote the quick dissociation of IGF1 from this intermediate ternary complex. These results suggest that after proteolysis of IGFBP3 CLD and CBP3 release (in the case of thrombin and ADAM12), IGF1 needs to be further dissociated from the destabilized IGF1/IGFBP3-NBP/ALS complex for IGF1R activation, which can be mediated by some other unknown factors at the extracellular matrix or plasma membrane. We have added these new data to the revised manuscript (p. 16-17, highlighted in yellow, and Fig. 6d, Supplementary Fig. 7b-7f and 7h-7j).

Supplementary Fig. 7 | f, Pull-down of the IGF1R ectodomain-leucine zipper-Strep (IGF1R-zip-strep) with Strep-Tactin resin after mixing IGF1R-zip-strep with IGF1-His or the intermediate ternary complex (IGF1-His/IGFBP3/ALS ternary complex after thrombin proteolysis). Immunoblot analysis was performed with anti-His and anti-Strep antibody to detect IGF1 bound to IGF1R-zip-strep and IGF1R-zip-strep bound to Strep-Tactin resin, respectively (left). Immunoblot analysis was also performed with anti-ALS, anti-IGFBP3, anti-His antibody to detect ALS, IGFBP3, and IGF1-His which are bound to IGF1R-zip-strep or are contained in flow through (right).

Discussion

The authors should emphasize that the cryo-EM deduced ALS structure clearly sustains the previous model proposed by David et al. as a flat horseshoe-like, and completely different to those proposed by Janosi et al. In addition, there are two important structural findings, reported in this manuscript, that were not observed in the David model: the long hook loop on LRRCT protruding into the central axis and the glycans attached to N368.

In response to these thoughtful suggestions, we have made revisions as follows (p. 18, highlighted in yellow):

“We also identified the important structural features of ALS such as the long hook loop on LRRCT and the glycans attached to N368, which have not been assessed by previous *in silico* structural model²⁰.”

Page 19

“It has long been known that IGFBP proteolysis of the linker domain leads to increased concentrations of bioavailable IGF and subsequent activation of the IGF1R³⁷. Surprisingly, we found the intermediate

ternary complex after proteolysis by thrombin or ADAM12, in which IGF1 is retained, but IGFBP3 CBP3 is released. This intermediate ternary complex can be explained by the previous results showing that CBPs have lower affinities than NBPs for IGF1^{35,36}.

The results obtained after proteolysis by thrombin and ADAM12 could not be generalized to other proteases, such as PAPP-A2. The authors should remark this aspect.

As mentioned above, we have added new results to the revised manuscript (p. 16-17, highlighted in yellow, and Fig. 6d and Supplementary Fig. 7b-7f and 7h-7j) and added the following sentence in the Discussion (p. 22, highlighted in yellow):

“Because of the different exclusive digestion sites, proteolysis with PAPP-A2 produced different fragments of IGFBP3 and the intermediate complex retaining CBP3. Moreover, the IGF1R activation potency of the PAPP-A2–treated ternary complex was less than those treated with thrombin and ADAM12, probably due to the limited proteolytic activity of PAPP-A2 against IGFBP3. However, our results suggest that proteolysis of the IGFBP3 CLD is required for IGF1R activation and that after the IGFBP3 CBP falls apart from the ternary complex by proteolysis (by thrombin or ADAM12), the remaining IGF1/IGFBP3-NBP/ALS complex will become relatively unstable, thereby further enhancing IGF1 bioavailability. We also postulate that the potential conformational changes of IGFBP3 CLD after proteolysis by thrombin or ADAM12 can cause subsequent dissociation of CBP from the ternary complex (Fig. 7c, bottom).”

Pages 19-20

“Given the crucial role of the IGF axis in growth and proliferation and the genetic defects in IGF, IGFBPs, and ALS associated with various human diseases, our study also can pave the way for identifying eventual therapeutic targets of IGF axis dysfunction.”

Since no genetic defect has yet been reported in any of the six IGFBPs, the IGFBPs should be omitted for the previous paragraph.

In response to these helpful suggestions, we have omitted “IGFBPs” in the revised manuscript (p. 23, highlighted in yellow)

Again, we thank all of the reviewers for their valuable assessments of our manuscript. We have carefully revised the manuscript based on their comments and hope that the revised manuscript is now acceptable for publication in *Nature Communications*.

REVIEWERS' COMMENTS

Reviewer #1 (Remarks to the Author):

The authors have done substantial work during the revision. They have satisfactorily address all my concerns raised on the initial manuscript. Overall, the entire manuscript has been improved significantly after the revision. This is a beautiful work, and important contribution to the IGF research. I support the publication of this work at Nature Communications.

Reviewer #2 (Remarks to the Author):

I am satisfied that the authors have addressed the reviewers' comments adequately and comprehensively. The changes have added significantly to the manuscript.

I have only two minor comments:

The table in Supp Fig 6 is difficult to understand. Would it make more sense if it were orientated with 5 columns each corresponding to a lane?

The revised sentence on page 12 of the rebuttal "IGF1 Y60H, which is highly conserved among species". Should read "IGF1 Y60, which is highly conserved among species".

Reviewer #3 (Remarks to the Author):

After revision this manuscript has been substantially improved.

The corrections and the set of experimental data, particularly those including site-directed mutagenesis of key amino acid residues in both the IGF1, and the ALS protein involved in crucial interface among IGF1/IGFBP3/ALS, have satisfactorily answered many of the concerns raised by the reviewers.

In my opinion in its present form this manuscript is now suitable for publication in the Nature Communication Journal.

This study represents an important contribution to our understanding of the complexity of both the assembly and the digestion of the IGF1/IGFBP3/ALS ternary complex. In addition, the spatial model of the ternary complex should be extremely useful to analyze the biological consequences of individual point mutations in the IGF1 and IGFALS genes (and eventually, in case of its description, point mutations in the IGFBP3 gene) on the formation and stability of ternary complexes.

Please check second paragraph in page 22:

In this sentence something is apparently missing.

It has long been known that IGFBP proteolysis of the linker domain leads to increased concentrations of bioavailable IGF and subsequent activation of the IGF1R48. Surprisingly, we found the intermediate ternary complex after proteolysis by thrombin and ADAM12, in which IGF1 is retained, but IGFBP3 CBP3 is released (?). These intermediate ternary complexes can be explained by the previous results showing that CBPs have lower affinities than NBPs for IGF146,47.

I think the authors are trying to say something like this:

It has long been known that IGFBP proteolysis of the linker domain leads to increased concentrations of bioavailable IGF and subsequent activation of the IGF1R48. Surprisingly, we found the intermediate ternary complex after proteolysis by thrombin and ADAM12, in which IGF1 is retained, but IGFBP3 CBP3 is released, can stimulate IGF1R phosphorylation. These intermediate ternary complexes can be explained by the previous results showing that CBPs have lower affinities than NBPs for IGF146,47.

Horacio M. Domené (PhD)

RESPONSE TO THE REVIEWERS' COMMENTS

We are grateful to the reviewers for their careful evaluation of our manuscript. In the text below, the reviewers' comments are in italics and our responses and descriptions of the changes made in the manuscript are in bold blue typeface.

Reviewer #1 (Remarks to the Author):

The authors have done substantial work during the revision. They have satisfactorily addressed all my concerns raised on the initial manuscript. Overall, the entire manuscript has been improved significantly after the revision. This is a beautiful work, and important contribution to the IGF research. I support the publication of this work at Nature Communications.

We appreciate these favorable and encouraging comments.

Reviewer #2 (Remarks to the Author):

I am satisfied that the authors have addressed the reviewers' comments adequately and comprehensively. The changes have added significantly to the manuscript.

We thank the reviewer for acknowledging the quality of our revision.

I have only two minor comments:

The table in Supp Fig 6 is difficult to understand. Would it make more sense if it were orientated with 5 columns each corresponding to a lane?

We appreciate the reviewer's helpful suggestion. We have revised Supplementary Fig. 6b (right, top), accordingly.

The revised sentence on page 12 of the rebuttal “IGF1 Y60H, which is highly conserved among species”. Should read “IGF1 Y60, which is highly conserved among species”.

We appreciate the reviewer’s careful inspection of our manuscript. We have revised this phrase as follows (p. 13, highlighted in character):

“IGF1 Y60, which is highly conserved among species and also conserved in IGF2 and insulin, is far away from the interaction interfaces of the ternary complex, whereas IGF1 R36 and V44 lie near the IGF1 interaction interface with ALS and CBP3, respectively. However, none of these mutations (R36Q, V44M, R50W, and Y60H) seem to affect IGF1 binding to ALS or IGFBP329, 30 (Fig. 4e).”

Reviewer #3 (Remarks to the Author):

After revision this manuscript has been substantially improved. The corrections and the set of experimental data, particularly those including site-directed mutagenesis of key amino acid residues in both the IGF1, and the ALS protein involved in crucial interface among IGF1/IGFBP3/ALS, have satisfactorily answered many of the concerns raised by the reviewers. In my opinion in its present form this manuscript is now suitable for publication in the Nature Communication Journal.

We thank the reviewer for acknowledging the quality of our revision.

This study represents an important contribution to our understanding of the complexity of both the assembly and the digestion of the IGF1/IGFBP3/ALS ternary complex. In addition, the spatial model of the ternary complex should be extremely useful to analyze the biological consequences of individual point mutations in the IGF1 and IGFALS genes (and eventually, in case of its description, point mutations in the IGFBP3 gene) on the formation and stability of ternary complexes.

Please check second paragraph in page 22:

In this sentence something is apparently missing.

It has long been known that IGFBP proteolysis of the linker domain leads to increased concentrations of bioavailable IGF and subsequent activation of the IGF1R48. Surprisingly, we found the intermediate ternary complex after proteolysis by thrombin and ADAM12, in which IGF1 is retained, but IGFBP3 CBP3 is released (?). These intermediate ternary complexes can be explained by the previous results showing that CBPs have lower affinities than NBPs for IGF146,47.

I think the authors are trying to say something like this:

It has long been known that IGFBP proteolysis of the linker domain leads to increased concentrations of bioavailable IGF and subsequent activation of the IGF1R48. Surprisingly, we found the intermediate ternary complex after proteolysis by thrombin and ADAM12, in which IGF1 is retained, but IGFBP3 CBP3

is released, can stimulate IGF1R phosphorylation. These intermediate ternary complexes can be explained by the previous results showing that CBPs have lower affinities than NBPs for IGF146,47.

Horacio M. Domené (PhD)

We have revised this phrase accordingly (p. 21, highlighted in blue character).